# Distributed Retraction-Free and Communication-Efficient Optimization on the Stiefel Manifold

Yilong Song [1]  Peijin Li [1]  Bin Gao [2]  Kun Yuan [1]

## Abstract

Optimization problems on the Stiefel manifold, ranging from principal component analysis to enhancing neural network robustness, are ubiquitous in machine learning. The Landing algorithm avoids computationally expensive retraction operations on manifolds, making it highly competitive for large-scale problems. This paper extends this method to distributed settings, introducing **EF-Landing**, the first retraction-free and communication-efficient algorithm for distributed stochastic optimization on the Stiefel manifold. By incorporating communication compression and error feedback, EF-Landing ensures convergence and constraint feasibility while significantly reducing communication overhead. We provide sharp convergence guarantees, demonstrating that EF-Landing achieves the same asymptotic linear speedup convergence rate as existing methods without communication compression. Furthermore, our analysis is highly versatile, applying to both deterministic and stochastic settings and encompassing algorithms based on gradient descent or momentum-based gradient descent. We also generalize EF-Landing to operate on blockwise Stiefel manifolds, enabling greater flexibility for structured constraints. Extensive numerical experiments validate our theoretical results.

## 1. Introduction

The Stiefel manifold, defined as the set of matrices with orthonormal columns, plays a crucial role in enhancing the dissimilarity between learned features in machine learning. Many classical problems inherently require orthogonal constraints, such as Principal Component Analysis (PCA) (Hotelling, 1933) and Canonical Correlation Analysis (CCA) (Hotelling, 1936). Furthermore, recent research has demonstrated that incorporating additional orthogonal constraints in deep learning problems often improves the robustness of neural networks (Arjovsky et al., 2016; Wang et al., 2020; Bansal et al., 2018). These insights underscore the importance of optimization on the Stiefel manifold in machine learning applications. Traditional methods typically involve a single computing node. However, state-of-the-art performance in modern tasks is often achieved using extremely large training datasets, necessitating efficient distributed algorithms for stochastic optimization on the Stiefel manifold across multiple computing nodes.

This paper considers the following optimization problem across $N$ collaborative nodes:

$$\min_{X \in \mathbb{R}^{n \times p}} f(X) = \frac{1}{N} \sum_{i=1}^{N} \big[ f_i(X) := \mathbb{E}_{\xi_i \sim \mathcal{D}_i} F(X; \xi_i) \big], \quad \text{(1a)}$$

$$\text{s.t.} \quad X^\top X = I_p, \quad \text{(1b)}$$

where $n \geq p$, $f_i(X)$ represents the objective function for each node $i$, and the random variable $\xi_i$ corresponds to the local data maintained by node $i$, following a local distribution $\mathcal{D}_i$. The constraint (1b) can also be expressed as $X \in \mathrm{St}(p, n)$ in which $\mathrm{St}(p, n) := \{X \in \mathbb{R}^{n \times p} \mid X^\top X = I_p\}$ is referred to as the Stiefel manifold. It is straightforward to verify that the Stiefel manifold constraint (1b) is non-convex.

Numerous approaches can be directly extended to solve the distributed problem (1). One line of research employs Riemannian methods to iteratively move towards the desired solution, incorporating retraction operations to ensure feasibility on the Stiefel manifold (Edelman et al., 1998; Absil et al., 2009; Absil & Malick, 2012). However, retraction operations are computationally expensive, often requiring matrix inversion, matrix exponential calculations, or QR factorization. To address this bottleneck, another line of research introduced the retraction-free Landing method (Ablin & Peyré, 2022), which relies solely on matrix multiplication and is particularly efficient on GPUs. In the Landing algorithm, iterates do not remain on the Stiefel manifold but gradually "land" (i.e., converge) onto it. This retraction-free property makes the method highly practical for large-scale

[1]Peking University [2]Academy of Mathematics and Systems Science, Chinese Academy of Sciences. Correspondence to: Kun Yuan <kunyuan@pku.edu.cn>.

*Proceedings of the 42$^{nd}$ International Conference on Machine Learning*, Vancouver, Canada. PMLR 267, 2025. Copyright 2025 by the author(s).

optimization problems. For these reasons, this paper focuses on developing and analyzing distributed Landing algorithms to solve problem (1).

In distributed optimization on the Stiefel manifold, each worker transmits gradient matrices to a central server to update the model parameters. Given the substantial size of these gradient matrices, communicating them at every iteration incurs significant overhead, which hinders algorithmic efficiency and scalability. To address this issue, this paper explores communication compression techniques (Alistarh et al., 2017; Richtarik et al., 2021; Stich et al., 2018; Huang et al., 2022) to reduce overhead. Instead of transmitting full gradient or model matrices, these strategies communicate compressed matrices with significantly smaller sizes at each iteration. While communication compression has demonstrated both theoretical guarantees and empirical successes in unconstrained distributed optimization, *no existing algorithms*, to the best of our knowledge, have been developed for distributed stochastic optimization on Stiefel manifolds. Several key questions arise when developing algorithms:

Q1. Which components of the algorithm can be compressed to ensure convergence? Specifically, should we compress the Euclidean gradient, Riemannian gradient, or gradient coupled with constraint penalty?

Q2. Is the error compensation strategy necessary for optimization on the Stiefel manifold to maintain both optimality and feasibility?

This paper addresses these questions and introduces the first retraction-free and communication-efficient algorithm for distributed stochastic optimization on the Stiefel manifold. Specifically, we make the following contributions:

- **EF-Landing algorithm.** We identify that compressing the Euclidean gradient of each $f_i(x)$ ensures both optimality and feasibility. Furthermore, we demonstrate that error feedback mechanisms are essential for convergence, even in single-node settings. Building on these insights, we propose the EF-Landing algorithm for distributed stochastic optimization problem (1) on the Stiefel manifold with communication compression.

- **Sharp convergence guarantees.** We provide convergence guarantees and establish convergence rates for EF-Landing. Our algorithm achieves the same asymptotic linear speedup convergence rate as the vanilla Landing algorithm without any communication compression. Furthermore, our results are highly versatile. By selecting appropriate hyper-parameters, our analysis applies to both deterministic and stochastic settings, encompassing a wide range of algorithms based on gradient descent or momentum-based gradient descent. Notably, our established convergence rates also recover results for communication compression in unconstrained distributed optimization. Our analysis is built upon a novel

merit function bound, which serves as a foundational tool for algorithms utilizing lossy gradient estimates and may be of independent interest.

- **Generalization to block-wise Stiefel manifolds.** We extend our framework to a generalized setting where matrix variables are partitioned into blocks with block-wise orthogonal constraints. This enables greater flexibility for large-scale optimization with structured constraints. We develop communication-efficient algorithms for this setting and provide convergence analysis, showing that the proposed methods retain the same theoretical guarantees as their non-block counterparts.

## 2. Related Work

**Optimization on manifolds.** Riemannian methods are classical approaches for solving optimization problems on manifolds. The foundational principles of Riemannian gradient descent with retraction were established in (Edelman et al., 1998; Absil et al., 2009; Absil & Malick, 2012). Building on gradient-based techniques, various algorithms and convergence results have been developed, including first-order methods (Zhang & Sra, 2016; Boumal et al., 2019), second-order methods (Absil et al., 2007; Qi et al., 2010), and accelerated methods (Liu et al., 2017; Ahn & Sra, 2020; Alimisis et al., 2021). Stochastic optimization on manifolds has also been explored (Bonnabel, 2013; Tripuraneni et al., 2018), with further refinements presented in (Zhang et al., 2016; Zhou et al., 2019). On the other hand, retraction-free approaches primarily include methods based on penalty functions and augmented Lagrangian formulations (Xiao et al., 2021; 2022; Gao et al., 2019), as well as techniques that reformulate manifold constraints (Lezcano-Casado & Martínez-Rubio, 2019; Liu et al., 2024). Additionally, the Landing method, designed specifically for the Stiefel manifold, has been proposed and analyzed in (Ablin & Peyré, 2022; Ablin et al., 2024; Vary et al., 2024).

**Distributed optimization.** Distributed optimization has been extensively studied, with Distributed Gradient Descent (Tsitsiklis et al., 1986) serving as the representative method. To address the communication bottleneck of central parameter servers, three main improvement strategies have been explored: decentralized communication (Lopes & Sayed, 2008; Shi et al., 2015; Nedić et al., 2017), lazy communication (McMahan et al., 2017; Stich, 2019a), and communication compression (Alistarh et al., 2017; Richtarik et al., 2021). Research on distributed optimization on manifolds remains relatively limited (Chen et al., 2021; Wang & Liu, 2022; Sun et al., 2024; Hu & Deng, 2024; Qu et al., 2024; Zhao et al., 2025; Zhang et al., 2024), including work such as (Sun et al., 2024) which incorporated the Landing algorithm with gradient tracking techniques, (Hu & Deng, 2024), which applied communication compression, and (Zhang

et al., 2024) which considered federated learning with local updates.

**Communication compression.** Communication compression significantly reduces the amount of information exchanged during distributed optimization. These techniques mainly fall two categories: sparsification methods, such as Rand-$K$ and Top-$K$ (Stich, 2019b; Wangni et al., 2018), and quantization methods, like QSGD and TurnGrad (Alistarh et al., 2017; Wen et al., 2017). To further mitigate information distortion, error feedback techniques have been widely adopted (Richtarik et al., 2021; Seide et al., 2014; Karimireddy et al., 2019; Stich & Karimireddy, 2021), with (Fatkhullin et al., 2023) showing that incorporating momentum enhances the effectiveness of error feedback. Optimal compelxity with communicaiton compression has been examined in (Huang et al., 2022; He et al., 2024).

## 3. Preliminary

**Notations.** Given a matrix $X \in \mathbb{R}^{m \times n}$, let $\|X\|_F$ denote the Frobenius norm, with the corresponding inner product defined as $\langle X, Y \rangle := \mathrm{Tr}(X^\top Y)$. For a square matrix $M \in \mathbb{R}^{n \times n}$, the symmetric and skew-symmetric components are given by $\mathrm{sym}(M) := \frac{1}{2}(M + M^\top)$ and $\mathrm{skew}(M) := \frac{1}{2}(M - M^\top)$, respectively.

### 3.1. Landing Method

**Descent direction.** The Landing method (Ablin & Peyré, 2022; Ablin et al., 2024) utilizes both the Riemannian gradient and a penalty term to enforce constraints while eliminating the need for retraction operation. The descent direction of Landing method is

$$\Lambda(X) = \mathrm{grad}f(X) + \lambda \nabla \mathcal{N}(X), \qquad (2)$$

$$\text{where} \quad \mathrm{grad}f(X) := \mathrm{skew}\Big(\nabla f(X)X^\top\Big)X,$$

$$\nabla \mathcal{N}(X) = X\Big(X^\top X - I_p\Big).$$

The first term, $\mathrm{grad}f(X)$, represents the Riemannian gradient on the Stiefel manifold $\mathrm{St}(p,n)$ with respect to the *canonical metric* (Gao et al., 2022), when $X$ satisfies the manifold constraints. The second term, $\nabla \mathcal{N}(X)$, denotes the gradient of the penalty term $\mathcal{N}(X) := \frac{1}{4}\|X^\top X - I_p\|_F^2$, where the penalty parameter $\lambda > 0$ controls its weight.

**Landing.** With descent direction (2), Landing method is:

$$X^{k+1} = X^k - \gamma \Lambda(X^k), \qquad (3)$$

where $\gamma$ is the step size. Rather than enforcing an exact constraint at each step, the Landing method only requires the iterations to remain within the neighborhood of $\mathrm{St}(p,n)$, i.e., the safe region defined as follows (Ablin et al., 2024):

**Definition 3.1** (Safe Region). Given some $\epsilon \in (0, 3/4)$, we define the safe region of a Stiefel manifold as

$$\mathrm{St}(p,n)^\epsilon := \Big\{ X \in \mathbb{R}^{n \times p} \mid \|X^\top X - I_p\| \leq \epsilon \Big\}.$$

The intuition behind the Landing algorithm is that $\mathrm{grad}f(X)$ drives the iteration towards the optimal solution on the Stiefel manifold, while $\nabla \mathcal{N}(X)$ steers the iteration towards the Stiefel manifold constraint. The orthogonality between these two terms ensures both optimality and feasibility as long as the algorithm converges.

### 3.2. Compressor and Error Feedback

**Contractive compressor.** This paper examines communication compression using contractive compressors, with Top-$K$ and Random-$K$ being commonly used examples (Wangni et al., 2018; Stich, 2019b).

**Definition 3.2** (Contractive Compressor). A compressor $\mathcal{C}$ is defined as a contractive compressor if it satisfies

$$\mathbb{E}_{\mathcal{C}}\Big[ \|\mathcal{C}(X) - X\|_F^2 \Big] \leq (1 - \alpha)\|X\|_F^2, \quad \forall X \in \mathbb{R}^{n \times p},$$

where $\alpha \in (0, 1]$ is the contractive factor. The expectation is taken over the randomness of the compression operator $\mathcal{C}$.

**Error feedback.** Consider the unconstrained problem:

$$\min_{X \in \mathbb{R}^{n \times p}} \quad f(X). \qquad (4)$$

The error feedback method (Richtarik et al., 2021) to solve the above unconstrained optimization problem is

$$X^{k+1} = X^k - \gamma Y^k, \qquad (5a)$$

$$Y^{k+1} = Y^k + \mathcal{C}(\nabla f(X^{k+1}) - Y^k), \qquad (5b)$$

where $Y^k$ is the compressed approximation of the gradient $\nabla f(X^k)$, and $\mathcal{C}(\cdot)$ satisfies Definition 3.2. The intuition behind error feedback is straightforward:

$$\mathbb{E}_{\mathcal{C}} \|Y^{k+1} - \nabla f(X^{k+1})\|^2$$
$$\overset{(5b)}{=} \mathbb{E}_{\mathcal{C}} \|Y^k - \nabla f(X^{k+1}) - \mathcal{C}(Y^k - \nabla f(X^{k+1}))\|^2$$
$$\leq (1 - \alpha)\|Y^k - \nabla f(X^{k+1})\|^2.$$

Suppose $\nabla f(X^k) \to \nabla f(X^*)$ as $k \to \infty$, where $X^*$ is a stationary solution to problem (4). The above inequality implies that $Y^k$ converges to $\nabla f(X^*)$. Combined with (5a), this guarantees that $X^k \to X^*$ under compression, thereby validating the effectiveness of the error feedback method.

## 4. EF-Landing Algorithm

In this section, we introduce **EF-Landing**, the first distributed **Landing** method incorporating **E**rror **F**eedback.

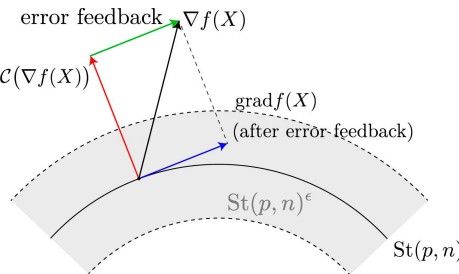

Figure 1: The necessity of error feedback

### 4.1. Compression on Euclidean Gradient

In the error feedback method for solving the unconstrained optimization problem (4), compressing the gradient $\nabla f(X)$ is sufficient to ensure convergence. Given that the Landing method can be expressed in a descent form (3), a natural question arises: should we mimic recursion (5) and compress $\Lambda(X)$ for EF-Landing?

We find that compressing $\Lambda(X)$ does not necessarily guarantee convergence. Note that the key factor ensuring the effectiveness of the Landing method (2)–(3) is the **orthogonality** property between the Riemannian gradient $\mathrm{grad} f(X)$ and the penalty gradient $\nabla\mathcal{N}(X)$, given by

$$\langle \mathrm{grad} f(X), \lambda\nabla\mathcal{N}(X)\rangle = 0, \quad \forall X \in \mathbb{R}^{n\times p}. \quad (6)$$

This orthogonality ensures that if $\|\Lambda(X)\|_F^2 \to 0$, then both $\|\mathrm{grad} f(X)\|_F^2 \to 0$ and $\|\lambda\nabla\mathcal{N}(X)\|_F^2 \to 0$ hold, guaranteeing optimality while preserving feasibility on the Stiefel manifold. However, it remains unclear whether orthogonality (6) is preserved after compressing $\Lambda(X)$.

To address this issue, we propose to compress the Euclidean gradient $\nabla f(X)$ in EF-Landing. Let $\boldsymbol{g}$ be the compressed approximation of the Euclidean gradient $\nabla f(X)$, we let

$$\tilde{\Lambda}(X; \boldsymbol{g}) = \mathrm{grad}(\boldsymbol{g}) + \lambda\nabla\mathcal{N}(X), \quad (7)$$

$$\text{where} \quad \mathrm{grad}(\boldsymbol{g}) := \mathrm{skew}\left(\boldsymbol{g}X^\top\right)X,$$

$$\nabla\mathcal{N}(X) = X\left(X^\top X - I_p\right).$$

The proposition below ensures the orthogonality between $\mathrm{grad}(\boldsymbol{g})$ and $\nabla\mathcal{N}(X)$ to hold after compressing $\nabla f(X)$:

**Proposition 4.1.** *For any gradient estimate $\boldsymbol{g} \in \mathbb{R}^{n\times p}$, the orthogonality between $\mathrm{grad}(\boldsymbol{g})$ and $\nabla\mathcal{N}(X)$ preserves (see proof in Appendix A.1):*

$$\langle \mathrm{grad}(\boldsymbol{g}), \nabla\mathcal{N}(X)\rangle = 0, \quad \forall X \in \mathbb{R}^{n\times p}, \forall \boldsymbol{g} \in \mathbb{R}^{n\times p}.$$

### 4.2. Error Feedback

**Vanilla compression may lead to non-convergence.** Various approaches exist for compressing the Euclidean gradient

$\nabla f(X)$. The most straightforward approach is to directly compress $\nabla f(X)$, i.e., $\boldsymbol{g} = \mathcal{C}(\nabla f(X))$, where $\mathcal{C}(\cdot)$ is the contractive compressor satisfying Definition 3.2. However, we observe that such vanilla compression may lead to non-convergence, even in the case of the single-node Landing (2)–(3). The main intuition is that the tangent space of the Stiefel manifold and the compressed gradient may become orthogonal, causing the iteration to stagnate. Figure 1 provides a schematic diagram of this phenomenon, while Proposition 4.2 provides the corresponding rigorous formulation.

**Proposition 4.2.** *There exists an L-smooth objective function $f : \mathbb{R}^{n\times p} \to \mathbb{R}$, a contractive compressor satisfying Definition 3.2, and an initial point $X^0 \in \mathbb{R}^{n\times p}$ such that the following update scheme:*

$$\boldsymbol{g} = \mathcal{C}(\nabla f(X)),$$
$$\tilde{\Lambda}(X; \boldsymbol{g}) = \mathrm{grad}(\boldsymbol{g}) + \lambda\nabla\mathcal{N}(X),$$
$$X \leftarrow X - \gamma\tilde{\Lambda}(X; \boldsymbol{g})$$

*results in the algorithm getting stagnant and failing to converge to the stationary solution. (See proof in Appendix A.2.)*

*Remark* 4.3. Proposition 4.2 suggests that vanilla gradient compression results in non-convergence for optimization on the Stiefel manifold, even in a deterministic and single-node setting. In contrast, for unconstrained deterministic optimization, vanilla gradient compression is guaranteed to converge under the same conditions (see proof in Appendix A.3). This highlights the additional challenges introduced by the Stiefel manifold constraint in algorithm design.

*Remark* 4.4. Error feedback technique can be motivated by diverse considerations. A similar work (Karimireddy et al., 2019) discussed the necessity of error feedback in the context of signSGD compressor (Bernstein et al., 2018). However, this analysis remains orthogonal to our scenario, as signSGD represents a specific type of compressor, and it can be proved that signSGD does not satisfy contractive compressor Definition 3.2. Consequently, the rationale for employing error feedback varies across different settings. Our theoretical analysis demonstrates that in our framework, it is precisely the orthogonal constraint that necessitates the use of error feedback.

**Error feedback corrects non-convergence.** Error feedback can correct the non-convergence encountered with vanilla gradient compression (more details in Appendix A.2). Inspired by momentum error feedback (Fatkhullin et al., 2023), we employ the following updates within each node $i$:

$$\boldsymbol{v}_i^{k+1} = (1-\eta)\boldsymbol{v}_i^k + \eta\nabla F(X^{k+1}; \xi_i^{k+1}), \quad (8a)$$
$$\boldsymbol{c}_i^k = \mathcal{C}(\boldsymbol{v}_i^{k+1} - \boldsymbol{g}_i^k), \quad (8b)$$
$$\boldsymbol{g}_i^{k+1} = \boldsymbol{g}_i^k + \boldsymbol{c}_i^k, \quad (8c)$$

**Algorithm 1** EF-Landing

**Require:** starting point $X^0 \in \mathbb{R}^{n \times p}$; gradient bound $L'$; step size $\gamma > 0$; compressor $\mathcal{C}$; momentum $\eta \in (0, 1]$;

1: Each node initializes $\boldsymbol{v}_i^0 = \nabla F(X^0; \xi_i^0)$, $\boldsymbol{g}_i^0 = \mathcal{C}(\boldsymbol{v}_i^0)$ for $i = 1, \ldots, N$; Master initializes $\boldsymbol{g}^0 = \frac{1}{N} \sum_{i=1}^N \boldsymbol{g}_i^0$.
2: **for** $k = 0, 1, \ldots, K - 1$ **do**
3:     Master clips the gradient via (10);
4:     Master computes $\tilde{\Lambda}(X^k; \tilde{\boldsymbol{g}}^k)$ using (7);
5:     Master computes $X^{k+1} = X^k - \gamma \tilde{\Lambda}(X^k; \tilde{\boldsymbol{g}}^k)$ and broadcasts $X^{k+1}$ to all nodes.
6:     **for** all nodes $i = 1, \ldots, N$ in parallel **do**
7:         Compute momentum $\boldsymbol{v}_i^{k+1}$ via (8a);
8:         Compress $\boldsymbol{c}_i^k$ via (8b) and send it to the master;
9:         Update local state $\boldsymbol{g}_i^{k+1}$ via (8c).
10:     **end for**
11:     Master updates $\boldsymbol{g}^{k+1}$ via (9).
12: **end for**

where $\eta$ is the momentum rate. Each node sends $\boldsymbol{c}_i^k$ to the master, and the master updates the Euclidean gradient:

$$\boldsymbol{g}^{k+1} = \boldsymbol{g}^k + \frac{1}{N} \sum_{i=1}^N \boldsymbol{c}_i^k. \qquad (9)$$

The compressed gradient $\boldsymbol{g}^k$ provides a better approximation than vanilla compressed gradient $\mathcal{C}(\nabla f(X^{k+1}))$ discussed in Section 4.2. Once $\boldsymbol{g}^k$ is obtained through the error feedback process (8)–(9), the master node will compute the descent direction $\tilde{\Lambda}(X; \boldsymbol{g})$ as outlined in (7).

### 4.3. EF-Landing Algorithm

EF-Landing comprises two key components: Euclidean gradient compression and error feedback, as outlined in the previous subsections. Additionally, before the master node computes the Landing descent direction (7), we apply a clipping operation to $\boldsymbol{g}^k$ to ensure that the integrated gradient estimate does not push the iterations beyond the safe region $\mathrm{St}(p, n)^\epsilon$ (see Definition 3.1). Given a gradient bound $L'$ (specified later in Section 5.1), we clip $\boldsymbol{g}^k$ as follows:

$$\tilde{\boldsymbol{g}}^k = \min\left\{1, \frac{L'}{\|\boldsymbol{g}^k\|_F}\right\} \boldsymbol{g}^k. \qquad (10)$$

Once clipped, $\boldsymbol{g}^k$ is ready for the computation of the Landing descent direction (7) and the iteration (3). The complete EF-Landing algorithm is presented in Algorithm 1.

The EF-Landing algorithm is highly versatile. When the gradient oracle satisfies $\nabla F(X; \xi_i) \equiv \nabla f_i(X)$, the problem reduces to deterministic optimization. The algorithm also applies when the compressor $\mathcal{C}(\cdot)$ is the identity mapping, corresponding to distributed optimization without communication compression. Furthermore, it remains valid when

the momentum rate is set to $\eta = 1$, representing algorithms without momentum. Notably, when all three conditions hold simultaneously, EF-Landing simplifies to the basic Landing algorithm with gradient clipping, with exact convergence results as established in (Ablin et al., 2024). Table 1 shows several scenarios to which EF-Landing is applicable.

Table 1: EF-Landing for diverse scenarios

| Scenarios | Deterministic | Stochastic (with momentum) |
|---|---|---|
| With compression | EF-Landing (Det.[♯]) $\sim \mathcal{O}(1/K)$ | EF-Landing (Sto.[♭]) $\sim \mathcal{O}(1/\sqrt{NK})$ |
| Without compression | Vanilla Landing $\sim \mathcal{O}(1/K)$ | Stochastic Landing $\sim \mathcal{O}(1/\sqrt{NK})$ |

[♯] Deterministic scenario. [♭] Stochastic scenario.

In the EF-Landing method (Algorithm 1), each node $i$ transmits the compressed variable $\boldsymbol{c}_i$ to the master node, while the master broadcasts the uncompressed variable $X$ to all workers. This unidirectional compression scheme is well-established in the literature (e.g., (Fatkhullin et al., 2023; Stich et al., 2018; Stich & Karimireddy, 2021; Alistarh et al., 2017)), as upload costs typically dominate download costs in distributed settings. Notably, the communication of $X$ can be removed in certain cases through system-level techniques such as the All-Reduce protocol (Patarasuk & Yuan, 2009), which eliminates the need for master-to-worker broadcasts.

## 5. EF-Landing Convergence Analysis

This section begins by presenting the necessary conditions, and then proceeds to derive the convergence guarantees and establish the convergence rates for EF-Landing.

### 5.1. Assumptions

The following assumptions are standard for optimization on the Stiefel manifold (Ablin et al., 2024) and distributed stochastic optimization.

**Assumption 5.1** (Global Gradient Bound). There exists constant $L'$ such that the following inequality holds

$$\|\nabla F(X; \xi_i)\|_F \leq L', \quad \forall X \in \mathrm{St}(p, n)^\epsilon, \forall \xi_i \sim \mathcal{D}_i.$$

*Remark* 5.2. The assumption of a gradient bound $L'$ is mild, as the search domain $\mathrm{St}(p, n)^\epsilon$ is bounded, and we assume that no highly anomalous data will cause gradient explosion.

**Assumption 5.3** (Lipschitz Smoothness). For each node $i$, $f_i(X) : \mathbb{R}^{n \times p} \to \mathbb{R}$ is differentiable and there exists a smooth constant $L_i$ such that for any $X, Y \in \mathrm{St}(p, n)^\epsilon$,

$$\|\nabla f_i(X) - \nabla f_i(Y)\|_F \leq L_i \|X - Y\|_F.$$

*Remark* 5.4. It is straightforward to show that the smoothness constant $L$ of the global function $f(X)$ satisfies $L \leq \max_{i=1,\ldots,N}\{L_i\}$. Additionally, we define $\tilde{L} := \sqrt{\frac{1}{N}\sum_{i=1}^{N} L_i^2}$ as the averaged smoothness constant.

**Assumption 5.5** (Lower Bounded Objective Function). The objective function $f(X)$ is lower bounded by $f^* := \inf_{X \in \text{St}(p,n)^\epsilon} f(X) > -\infty$.

**Assumption 5.6** (Unbiasedness and Bounded Variance). There exists a $\sigma \geq 0$ such that for each node $i$, for any $X \in \text{St}(p,n)^\epsilon$, it holds that

$$\mathbb{E}_{\xi_i \sim \mathcal{D}_i}[\nabla F(X;\xi_i) - \nabla f_i(X)] = 0,$$
$$\mathbb{E}_{\xi_i \sim \mathcal{D}_i}\left[\|\nabla F(X;\xi_i) - \nabla f_i(X)\|_F^2\right] \leq \sigma^2.$$

### 5.2. Supporting Lemmas

This section establishes supporting lemmas for the convergence analysis of the EF-Landing algorithm.

**Safe step size.** The main objective of the Landing algorithm is to select an appropriate step size that ensures the iterations remain within $\text{St}(p,n)^\epsilon$. In our analysis, we require only two assumptions—Assumptions 5.1 and 5.3—to determine a uniform, non-vanishing step size that is valid throughout the entire iteration process.

**Lemma 5.7** (Uniform Safe Step Size). *Consider the update $\tilde{X} = X - \gamma\tilde{\Lambda}(X;g)$, where $\tilde{\Lambda}(X;g)$ is defined in* (7) *and $\gamma$ is a step size. For any $X \in \text{St}(p,n)^\epsilon$, $g \in \mathbb{R}^{n \times p}$, if $\|g\|_F \leq L'$ and we choose*

$$\gamma \leq \gamma_s := \min\left\{\frac{\lambda(1-\epsilon)\epsilon}{(1+\epsilon)^2(L')^2 + \lambda^2(1+\epsilon)\epsilon^2},\right.$$
$$\left.\sqrt{\frac{\epsilon}{2(1+\epsilon)^2(L')^2}}, \frac{1}{2\lambda}\right\},$$

*then the next iteration $\tilde{X}$ remains in $\text{St}(p,n)^\epsilon$ (see proof in Appendix B.1).*

**Merit function.** We introduce a merit function to facilitate convergence analysis of EF-Landing:

$$m(X) = f(X) - h(X) + \mu\mathcal{N}(X), \tag{11}$$
$$\text{where } h(x) = \frac{1}{2}\langle \text{sym}(X^\top\nabla f(X)), X^\top X - I_p\rangle.$$

The constant $\mu$ is a hyperparameter, and its value will be specified in Lemma 5.8. The merit function $m(X)$ is $L_m$-smooth when restricted to the bounded safe region $\text{St}(p,n)^\epsilon$, with an upper bound for $L_m$ provided by (Ablin et al., 2024).

Furthermore, under Assumption 5.5, since both $h(X)$ and $\mathcal{N}(X)$ are bounded for $X \in \text{St}(p,n)^\epsilon$, the merit function $m(X)$ is also lower-bounded, with its infimum denoted as $m^*$. In this work, we derive a lower bound for

$\langle\nabla m(X), \tilde{\Lambda}(X;g)\rangle$. This bound plays a critical role in analyzing the impact of compression error, as it explicitly isolates the difference term $\|g - \nabla f(X)\|_F^2$.

**Lemma 5.8** (Merit Function Bound). *For all $X \in \text{St}(p,n)^\epsilon$ and $g \in \mathbb{R}^{n \times p}$, the inner product between the descent direction* (7) *and the gradient of the merit function* (11) *satisfies the lower bound (see proof in Appendix B.2)*

$$\langle\tilde{\Lambda}(X;g), \nabla m(X)\rangle \geq \frac{1}{4}\|\text{grad}f(X)\|_F^2 + \lambda\mu\mathcal{N}(X)$$
$$+ \frac{1}{4}\|\text{skew}(gX^\top)X\|_F^2 - \|g - \nabla f(X)\|_F^2,$$

*provided that $\mu \geq \frac{2}{3-4\epsilon}\left(L(1-\epsilon) + 3\sqrt{1+\epsilon}L' + 2\hat{L}^2\frac{1+\epsilon}{\lambda}\right)$, where $L$ and $L'$ are defined in Assumptions 5.3 and 5.1, respectively, $\hat{L} = \max\{L, L'\}$, and $\epsilon < 3/4$.*

### 5.3. Main Convergence Theorem

Given an arbitrary momentum coefficient $\eta > 0$, we define a Lyapunov function as follows:

$$\mathcal{L}^k := m(X^k) - m^* + \frac{c_1\gamma}{\theta}\tilde{G}^k + \frac{2c_1\gamma\eta\beta}{\theta}\tilde{P}^k + \frac{c_2\gamma}{\eta}P^k \tag{12}$$

$$\text{where} \quad \tilde{G}^k := \frac{1}{N}\sum_{i=1}^{N}\|g_i^k - v_i^k\|_F^2,$$
$$\tilde{P}^k := \frac{1}{N}\sum_{i=1}^{N}\|v_i^k - \nabla f_i(X^k)\|_F^2,$$
$$P^k := \|v^k - \nabla f(X^k)\|_F^2,$$

where $v^k := \frac{1}{N}\sum_{i=1}^{N} v_i^k$, and $m^*$ denotes the infimum of the merit function in (11). Constants $\theta, \beta, c_1, c_2$ will be determined later. With the above Lyapunov function, we derive the main convergence theorem as follows.

**Theorem 5.9** (Main Convergence Theorem). *Letting Assumptions 5.1, 5.3, 5.5 and 5.6 hold, if compressor $\mathcal{C}$ satisfies Definition 3.2, for an arbitrary momentum coefficient $\eta \in (0,1]$, by running Algorithm 1 for $K$ iterations, with proper learning rate $\gamma$ (see Appendix B.3), we have*

$$\frac{1}{4K}\sum_{k=0}^{K-1}\mathbb{E}[\|\text{grad}f(X^k)\|_F^2] + \frac{\lambda\mu}{2K}\sum_{k=0}^{K-1}\mathbb{E}[\mathcal{N}(X^k)]$$
$$\leq \frac{\mathcal{L}^0}{\gamma K} + \frac{c_1\eta^2(1-\alpha)\sigma^2}{\theta} + \frac{2c_1\eta^3\beta\sigma^2}{\theta} + \frac{c_2\eta\sigma^2}{N},$$

*In the inequality, $\mathcal{L}^k$ is defined in* (12), *$\theta$ and $\beta$ are two scalars defined as $\theta := 1 - \sqrt{1-\alpha}$ and $\beta := \frac{1-\alpha}{1-\sqrt{1-\alpha}}$. $c_1, c_2$ are two constants differently specified in three cases:*

- *When the momentum coefficient $\eta = 1$ and the variance of gradients $\sigma^2 = 0$, the momentum and stochastic error is 0, and $c_1, c_2$ are specified as $c_1 = 1, c_2 = 0$.*
- *When the contractive factor $\alpha = 1$, the compression error is 0, and $c_1, c_2$ are specified as $c_1 = 0, c_2 = 1$.*
- *Otherwise, we specify $c_1 = c_2 = 2$.*

*Remark* 5.10. The convergence theorem above is highly versatile. By appropriately selecting hyper-parameters such as the momentum coefficient $\eta$, gradient variance $\sigma$, and compressor factor $\alpha$, our analysis applies to both deterministic and stochastic settings, covering algorithms based on gradient descent and momentum-based gradient descent, with or without communication compression.

### 5.4. Convergence Rate in Deterministic Scenario

In this subsection, we set $\sigma^2 = 0$ in Assumption 5.6. We further set $\eta = 1$ to completely eliminate both momentum and stochastic error. Under these conditions, the constants in Theorem 5.9 simplify to $c_1 = 1$ and $c_2 = 0$, allowing the theorem to naturally recover various convergence results in deterministic scenarios.

**Theorem 5.11** (EF-Landing in Deterministic Scenarios). *Letting Assumptions 5.1, 5.3, 5.5 and 5.6 hold, if compressor $\mathcal{C}$ satisfies Definition 3.2, $\sigma^2 = 0$, and $\eta = 1$, by running Algorithm 1 for $K$ iterations, with proper constant learning rate $\gamma$ (see Appendix B.4), we have*

$$\frac{1}{K}\sum_{k=0}^{K-1}\mathbb{E}[\|\mathrm{grad}f(X^k)\|_F^2] \leq \frac{4(m(X^0)-m^*)}{K\gamma} + \frac{4\mathbb{E}[\tilde{G}^0]}{K\theta},$$

$$\frac{1}{K}\sum_{k=0}^{K-1}\mathbb{E}[\mathcal{N}(X^k)] \leq \frac{2(m(X^0)-m^*)}{K\gamma\lambda\mu} + \frac{2\mathbb{E}[\tilde{G}^0]}{K\theta\lambda\mu},$$

*where $\theta$ and $\beta$ are defined in Theorem 5.9.*

By additionally setting $\alpha = 1$ and $\tilde{G}^0 = 0$, meaning no compression is applied to the Landing method, the theorem simplifies to the standard Landing convergence theorem:

**Corollary 5.12** (Vanilla Landing in Deterministic Scenarios). *Letting Assumptions 5.1, 5.3, 5.5 and 5.6 hold, if compressor $\mathcal{C}$ satisfies Definition 3.2, $\sigma^2 = 0$, $\eta = 1$ and $\alpha = 1$, by running Algorithm 1 for $K$ iterations, with proper constant learning rate $\gamma$ (see Appendix B.4), we have*

$$\frac{1}{K}\sum_{k=0}^{K-1}\|\mathrm{grad}f(X^k)\|_F^2 \leq \frac{4(m(X^0)-m^*)}{K\gamma},$$

$$\frac{1}{K}\sum_{k=0}^{K-1}\mathcal{N}(X^k) \leq \frac{2(m(X^0)-m^*)}{K\gamma\lambda\mu}.$$

*Remark* 5.13. Our established rate for vanilla Landing without compression is $\mathcal{O}(1/K)$, matching the state-of-the-art result in (Ablin et al., 2024; Vary et al., 2024) with exact constants, confirming the sharpness of our analysis. Moreover, by comparing Theorem 5.11 and Corollary 5.12, we observe that Landing, with or without communication compression, achieves the same $\mathcal{O}(1/K)$ rate, indicating that compression does not degrade the convergence order. However, more aggressive compression (i.e., $\theta \to 0$) leads to slower convergence for EF-Landing.

### 5.5. Convergence Rate in Stochastic Scenario

In this section, we consider the case where the variance $\sigma^2 > 0$ (Assumption 5.6). Notably, our assumption differs from that of (Ablin et al., 2024), which directly assumes a bounded variance for $\tilde{\Lambda}(X^k; \tilde{\boldsymbol{g}}^k)$. The momentum technique plays a crucial role in stochastic settings.

We consider the case with communication compression, i.e., $\alpha < 1$, and analyze the most general version of the EF-Landing algorithm. In this setting, the constants in Theorem 5.9 are specified as $c_1 = c_2 = 2$. The following theorem presents the convergence result of EF-Landing in stochastic scenarios, demonstrating a linear speedup.

**Theorem 5.14** (EF-Landing in Stochastic Scenarios). *Letting Assumptions 5.1, 5.3, 5.5 and 5.6 hold, if compressor $\mathcal{C}$ satisfies Definition 3.2, and we choose step size $\gamma$ as in Lemma B.6, by running Algorithm 1 for $K$ iterations, with proper learning rate $\gamma$ and momentum coefficient $\eta$ (see Appendix B.6), we have*

$$\frac{1}{K}\sum_{k=0}^{K-1}\mathbb{E}[\|\mathrm{grad}f(X^k)\|_F^2] \leq 4\mathcal{O}\Big(\frac{1}{\sqrt{NK}}\Big),$$

$$\frac{1}{K}\sum_{k=0}^{K-1}\mathbb{E}[\mathcal{N}(X^k)] \leq \frac{2}{\lambda\mu}\mathcal{O}\Big(\frac{1}{\sqrt{NK}}\Big),$$

*where the $\mathcal{O}(\frac{1}{\sqrt{NK}})$ term is specified as*

$$\mathcal{O}\Big(\frac{1}{\sqrt{NK}}\Big) = \frac{C_1^\gamma \mathcal{L}^0}{K} + 4\Big(\frac{2\sigma^2 C_2^\gamma \mathcal{L}^0}{NK}\Big)^{\frac{1}{2}}$$

$$+ 4\Big(\frac{2\sigma^2(1-\alpha)}{\theta}\Big)^{\frac{1}{3}}\Big(\frac{C_2^\gamma \mathcal{L}^0}{K}\Big)^{\frac{2}{3}} + 4\Big(\frac{4\beta\sigma^2}{\theta}\Big)^{\frac{1}{4}}\Big(\frac{C_2^\gamma \mathcal{L}^0}{K}\Big)^{\frac{3}{4}},$$

*The constant $\mathcal{L}^0$ is defined in (12), while $\theta$ and $\beta$ are specified in Theorem 5.9. The constants $C_1^\gamma$ and $C_2^\gamma$ are independent of $N$ and $K$, with their definitions provided in Appendix B.6.*

*Remark* 5.15. This theorem establishes that EF-Landing achieves an asymptotic linear speedup convergence rate. The impact of the communication compressor appears only in higher-order terms and does not affect the dominant linear speedup rate. Notably, this rate aligns with the momentum error feedback method (Fatkhullin et al., 2023), which is designed for unconstrained minimization problems. This suggests that, with careful algorithmic design, the Stiefel manifold constraint does not pose fundamental challenges to communication compression.

To illustrate the versatility of our convergence analysis, we now examine the stochastic Landing algorithm without compression. In this case, the constants in Theorem 5.9 are specified as $c_1 = 0$ and $c_2 = 1$, leading to the result:

**Theorem 5.16** (Stochastic Landing Convergence). *Letting Assumptions, 5.1, 5.3, 5.5 and 5.6 hold, if compressor $\mathcal{C}$*

*satisfies Definition 3.2, $\alpha = 1$, with proper learning rate $\gamma$ and momentum coefficient $\eta$ (see Appendix B.7), by running Algorithm 1 for $K$ iterations, we have*

$$\frac{1}{K} \sum_{k=0}^{K-1} \mathbb{E}[\|\mathrm{grad}f(X^k)\|_F^2] \leq \frac{4C_1^\gamma \mathcal{L}^0}{K} + 4\sigma\sqrt{\frac{C_2^\gamma \mathcal{L}^0}{NK}},$$

$$\frac{1}{K} \sum_{k=0}^{K-1} \mathcal{N}(X^k) \leq \frac{2C_1^\gamma \mathcal{L}^0}{K\lambda\mu} + \frac{2\sigma}{\lambda\mu}\sqrt{\frac{C_2^\gamma \mathcal{L}^0}{NK}},$$

*where $\mathcal{L}^0$ is defined in (12), $\theta$ and $\beta$ are defined in Theorem 5.9, $C_1^\gamma, C_2^\gamma$ are two constants defined in Appendix B.7.*

*Remark* 5.17. Our analysis demonstrates that the vanilla Landing method achieves a convergence rate of $\mathcal{O}(1/\sqrt{NK})$ in stochastic scenarios without compression. Furthermore, by comparing Theorem 5.14 and Theorem 5.16, we note that the Landing method attains the same $\mathcal{O}(1/\sqrt{NK})$ rate regardless of whether communication compression is applied. This indicates that the use of compression does not adversely affect the convergence order.

# 6. Optimization on Block-wise Manifolds

Practical problems typically require variables to satisfy orthogonal constraints in a block-wise manner, while the remaining variables remain unconstrained, e.g., orthogonal CNN with orthogonally regularized convolutional layers and unconstrained MLP layers (Wang et al., 2020). The optimization problem can thus be formulated as follows:

$$\min_{X_1,\ldots,X_J;x} \quad f(X_1,\ldots,X_J;x) \tag{13}$$
$$\mathrm{s.\,t.} \quad X_j^\top X_j = I_{p_j}, \quad j = 1,\ldots,J,$$

where $X_j \in \mathbb{R}^{n_j \times p_j}$ represents a constrained variable block on $\mathrm{St}(p_j, n_j)$, and $x \in \mathbb{R}^{n_0}$ contains the remaining unconstrained variables. We define $\mathfrak{X} := (X_1,\ldots,X_J;x)$, a composite data type, whose domain is given by $\mathfrak{R} := (\mathbb{R}^{n_1 \times p_1}, \ldots, \mathbb{R}^{n_J \times p_J}; \mathbb{R}^{n_0})$. Consequently, $f$ is well-defined as a mapping from $\mathfrak{R}$ to $\mathbb{R}$, with its gradient as

$$\nabla f(\mathfrak{X}) := \Big(\frac{\partial f}{\partial X_1}, \ldots, \frac{\partial f}{\partial X_J}; \frac{\partial f}{\partial x}\Big) \in \mathfrak{R}.$$

The inner product on $\mathfrak{R}$ is computed as

$$\langle \mathfrak{X}, \mathfrak{Y} \rangle := \sum_{j=1}^{J} \langle X_j, Y_j \rangle + \langle x, y \rangle, \quad \mathfrak{X}, \mathfrak{Y} \in \mathfrak{R}.$$

The norm on $\mathfrak{R}$ is defined as $\|\mathfrak{X}\| := \sqrt{\langle \mathfrak{X}, \mathfrak{X} \rangle}$, allowing Definition 3.2, Assumptions 5.1, 5.3, 5.5, and 5.6 to be properly formulated for problem (13).

By applying the Landing descent direction to each orthogonally constrained block and vanilla gradient descent to the free variable block, the descent direction of the block-wise constrained problem (13) can be expressed as:

$$\tilde{\Lambda}(\mathfrak{X}; \mathfrak{g}) := \Big(\mathrm{skew}(\boldsymbol{g}_1 X_1^\top)X_1 + \lambda X_1(X_1^\top X_1 - I_{p_1}),$$
$$\vdots$$
$$\mathrm{skew}(\boldsymbol{g}_J X_J^\top)X_J + \lambda X_J(X_J^\top X_J - I_{p_J});$$
$$\boldsymbol{g}_0\Big) \in \mathfrak{R}, \tag{14}$$

where $\boldsymbol{g}_1, \ldots, \boldsymbol{g}_J, \boldsymbol{g}_0$ can be $\partial f/\partial X_1, \ldots, \partial f/\partial X_J$, $\partial f/\partial x$ or their compressed approximations, as indicated by (8)–(9). Using (14), the main update step follows:

$$\mathfrak{X}^{k+1} = \mathfrak{X}^k - \gamma\tilde{\Lambda}(\mathfrak{X}; \mathfrak{g}). \tag{15}$$

Therefore, it is straightforward to extend the EF-Landing Algorithm 1 to solve the optimization problem (13) on the block-wise Stiefel manifolds. The implementation details are provided in Appendix C.1. Moreover, Appendix C.2 presents the convergence results of EF-Landing in stochastic scenario with block-wise constraints.

# 7. Numerical Experiments

To validate the performance of EF-Landing, we provide experiments on two groups of problems: the distributed online PCA for deterministic scenario and deep learning using ResNet-18 (He et al., 2016) neural network architecture with orthogonal constraints applied to the convolutional layers for stochastic scenario. We compared EF-Landing with other algorithms for optimization on the Stiefel manifold, including vanilla Landing, QR retraction and the Euclidean gradient descent with added $\ell_2$ squared penalty norm. In addition, to ensure the sufficiency and rigor of the experiments, we compared our approach with several other distributed algorithms on manifolds, including decentralized training frameworks (Chen et al., 2021) and centralized frameworks with multiple local updates before aggregation (Zhang et al., 2024). In all experiments, the gradient clipping bound $L'$ was set to a sufficiently large value, as overestimating it does not incur any loss. All the experiments were implemented in PyTorch and performed using a single GPU. Further experiments can be found in Appendix D.

## 7.1. Distributed Online PCA

The distributed online PCA problem can be expressed as

$$\min_{X \in \mathrm{St}(p,n)} \quad f(X) := -\frac{1}{2N} \sum_{i=1}^{N} \|A_i X\|_F^2,$$

where $A_i \in \mathbb{R}^{l \times n}$ is the synthetically generated local data matrix for node $i$, with $l$ being the number of samples for each node. We set $l = 5000$, $n = 5000$, $N = 4$ and

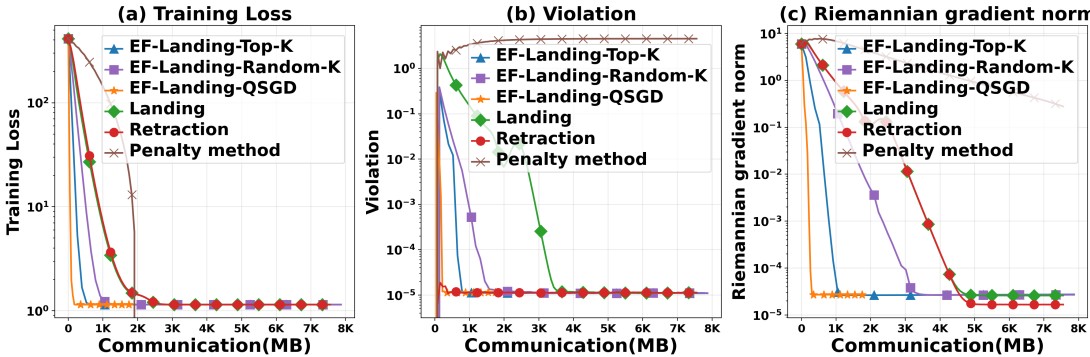

Figure 2: Performance comparison of EF-Landing and other algorithms on distributed online PCA with $p = 1000$.

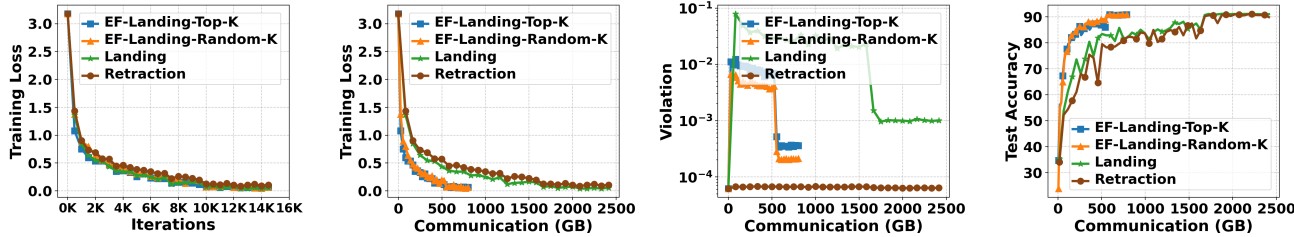

Figure 3: Performance comparison on deep learning with ResNet-18 on CIFAR-10.

$p = 1000$. For EF-Landing algorithm, the penalty parameter $\lambda$ was set to 1, and we used three compressors: Top-$K$, Rand-$K$ with compression retention ratio 0.1 and QSGD with quantization level $s = 16$. Using different algorithms mentioned above, we get the results as in Figure 2.

In the experiments, the training loss was computed by $f(X_k) - f(X_*)$, where $X_*$ is the matrix composed by $p$ right singular vectors of $A = \left(A_1^\top, \cdots, A_N^\top\right)^\top$, and the violation of constraint was computed by $\mathcal{N}(X) = \frac{1}{4}\|X^\top X - I_p\|^2$. The result shows that EF-Landing reaches the same training loss as the vanilla Landing algorithm and other algorithms. Furthermore, due to communication compression technique, EF-Landing significantly reduces the communication overhead. For penalty method, the penalty parameter was set as $\lambda = 8$. We observe that the penalty method performs poorly on this problem, which could not enforce the orthogonal constraint during the optimization process (its negative part of training loss was omitted). We provide further discussion on the choice of penalty parameters in Appendix D.3.

### 7.2. Neural Network with Orthogonal Constraints

Another group of experiments were conducted on Resnet18. We apply orthogonal constraints to the convolutional layers by reshaping the convolutional kernels to size $n_{\text{out}} \times n_{\text{in}}n_xn_y$, where $n_{\text{in}}$ and $n_{\text{out}}$ are the numbers of input and output channels, and $n_x, n_y$ are the filter dimensions (Ablin

et al., 2024). In case that the reshaped matrix becomes wide instead of tall, we enforce orthogonality on its transpose. Problem formulation (13) is suitable for our settings.

We tested the performance of EF-Landing on the CIFAR-10 dataset, using Top-$K$ and Random-$K$ compressors with a compression retention ratio of 0.2. We compared the results of EF-Landing with vanilla Landing and QR retraction methods in Figure 3. For each algorithm, the network is trained for 150 epochs, and the learning rate is reduced to 0.1 of its original value after the 100th epoch.

As shown in Figure 3, EF-Landing and other algorithms reach similar accuracy in terms of iterations, while EF-Landing significantly reduces the communication overhead. Additional experimental details, including the choice of hyper-parameters such as momentum and step size, can be found in Appendix D.

## 8. Conclusion

In this paper, we propose EF-Landing algorithm for solving distributed optimization problems on Stiefel manifolds, which is both computationally and communicationally efficient. EF-Landing encompasses various scenarios with sharp convergence guarantees and linear speedup rate. The good compatibility with block-wise problems extends the practicality of EF-Landing even further. Extensive numerical experiments validate our theoretical results.

## Acknowledgements

This work was supported by the National Natural Science Foundation of China under Grants 92370121, 12301392, and W2441021, and by the National Key Research and Development Program of China under Grant 2024YFA1012902. We also thank the anonymous reviewers for their valuable feedback.

## Impact Statement

This paper centers on the theoretical analysis of machine learning algorithm convergence. We do not foresee any significant societal consequences arising from our work, none of which we believe need to be explicitly emphasized.

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

## Contents of the Appendices

## A. Details of Realizing EF-Landing

### A.1. Proof of Proposition 4.1

*Proof.* For any $g \in \mathbb{R}^{n \times p}$, we have

$$\langle \mathrm{skew}(gX^\top)X, \lambda X(X^\top X - I_p)\rangle = \mathrm{Tr}\big(X^\top \mathrm{skew}(gX^\top)^\top \cdot \lambda X(X^\top X - I_p)\big)$$
$$= \mathrm{Tr}\big(\mathrm{skew}(gX^\top)^\top \cdot \lambda X(X^\top X - I_p)X^\top\big) = \langle \mathrm{skew}(gX^\top), \lambda X(X^\top X - I_p)X^\top\rangle,$$

where $\mathrm{skew}(gX^\top)$ is a skew-symmetric matrix and $\lambda X(X^\top X - I_p)X^\top$ is a symmetric one. For any skew-symmetric matrix $X$ and symmetric matrix $Y$,

$$\langle X, Y\rangle = \langle X^\top, Y^\top\rangle = -\langle X, Y\rangle = 0,$$

which leads to the conclusion. $\square$

### A.2. Example Showing the Necessity of Error Feedback

Here is a toy example illustrating the phenomenon of stagnation. Let $X \in \mathbb{R}^{2 \times 1}$ and the Stiefel manifold $\{X \in \mathbb{R}^{2 \times 1} \mid X^\top X = 1\}$ is now a unit circle. We only consider distributed learning on a single node with gradient $\nabla f(X) = (2,1)^\top$. Top-1 compressor will compress the gradient into $\mathcal{C}(\nabla f(X)) = (2,0)^\top$. If the iteration point $X$ happens to be $(1,0)^\top$, the Riemannian gradient after compression $\mathrm{grad}(\boldsymbol{g}) = \mathrm{skew}(\boldsymbol{g}X^\top)X$ will be $\mathbf{0}$ since $\mathcal{C}(\nabla f(X)) = (2,0)^\top$ happens to be orthogonal to the tangent space of Stiefel manifold at point $X = (1,0)^\top$, i.e., $\mathrm{span}\{(0,1)^\top\}$. The iteration point $X$ is on $\mathrm{St}(1,2)$, so the gradient of penalty term $\lambda \mathcal{N}(X)$ is also $\mathbf{0}$. Therefore, the descent direction $\hat{\Lambda}(X; \mathcal{C}(\nabla f(X)))$ is $\mathbf{0}$ but the iteration does not reach a stationary point. Furthermore, greedy compressor Top-1 will permanently output the same compressed gradient $(2,0)^\top$ as long as the iteration point does not move. This causes the iteration to stagnate.

However, when using error feedback, for the same situation, we first make a stagnant step to $X' = X = (1,0)^\top$. But since the error feedback technique compresses the difference of two consecutive gradients, the next gradient after compression will be $\mathcal{C}(\nabla f(X)) + \mathcal{C}(\nabla f(X') - \mathcal{C}(\nabla f(X))) = (2,0)^\top + (0,1)^\top = (2,1)^\top$, which is no longer orthogonal to the tangent space of $X'$ on the Stiefel manifold because the direction parallel with the tangent space comes into effect. Hence, the iteration escapes the stagnation point. Figure 4 illustrates the process.

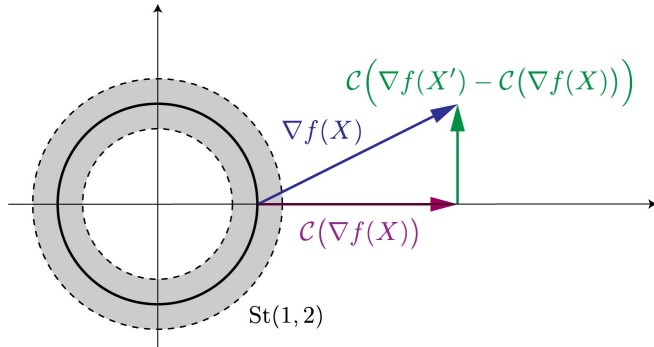

Figure 4: Diagram for the toy example in Appendix A.2.

The heuristic reason explaining the phenomenon of stagnation is: Greedy compressors choose the directions with largest magnitude in unconstrained Euclidean space, but these directions are not necessarily directions with largest magnitude in feasible subspaces of the Stiefel manifold. In extreme cases, when the two sets of directions do not intersect at all and the greedy compressors cannot give other feasible directions for the current iteration point, the iteration falls into stagnation. Error feedback, however, tends to select directions that were discarded in the previous iteration, ensuring that the feasible directions of the Stiefel manifold could always be selected within a few steps. Hence, error feedback technique overcomes the phenomenon of stagnation. In fact, for any deterministic optimization problem on the Stiefel manifold, when using greedy compressors, stagnation occurs with high probability. Thus this is a general problem that is eventually solved by error feedback technique.

### A.3. Unconstrained Deterministic Optimization with Vanilla Gradient Compression

The error feedback strategy was introduced in (Richtarik et al., 2021) mainly to eliminate the instability caused by compression among more than one nodes at the stationary point. A straightforward idea is that when using contractive compressors, the same issue does not exist in the single-node scenario without any extra constraint. In fact, we have the following algorithm and its convergence result.

---

**Algorithm 2** Unconstrained Deterministic Optimization with Vanilla Gradient Compression (Single node)

---

1: **Input:** starting point $x^0 \in \mathbb{R}^d$, learning rate $\gamma > 0$
2: **for** $t = 0, 1, 2, \ldots, T-1$ **do**
3:     $g^k = \mathcal{C}(\nabla f(x^k))$
4:     $x^{k+1} = x^k - \gamma g^k$
5: **end for**

---

**Theorem A.1.** *Suppose $f : \mathbb{R}^d \to \mathbb{R}$ is $L$-smooth and lower bounded by $f^*$, and the compressor $\mathcal{C}$ satisfies Definition 3.2, if we choose the step size $\gamma < 1/L$, by running Algorithm 2 for $K$ iterations, we have*

$$\frac{1}{K}\sum_{k=0}^{K-1} \mathbb{E}[\|\nabla f(x^k)\|^2] \leq \frac{2(f(x^0) - f^*)}{\gamma \alpha K}$$

*Proof.* Using $L$-smoothness of $f$, we have

$$
\begin{aligned}
f(x^{k+1}) &\leq f(x^k) + \langle \nabla f(x^k), x^{k+1} - x^k \rangle + \frac{L}{2}\|x^{k+1} - x^k\|^2 \\
&= f(x^k) - \gamma \langle \nabla f(x^k), g^k \rangle + \frac{L}{2}\|x^{k+1} - x^k\|^2 \\
&= f(x^k) - \frac{\gamma}{2}\|\nabla f(x^k)\|^2 - \frac{\gamma}{2}\|g^k\|^2 + \frac{\gamma}{2}\|g^k - \nabla f(x^k)\|^2 + \frac{L}{2}\|x^{k+1} - x^k\|^2 \\
&= f(x^k) - \frac{\gamma}{2}\|\nabla f(x^k)\|^2 - (\frac{1}{2\gamma} - \frac{L}{2})\|x^{k+1} - x^k\|^2 + \frac{\gamma}{2}\|g^k - \nabla f(x^k)\|^2.
\end{aligned}
$$

Noticing that Definition 3.2 of contractive compressor leads to

$$\mathbb{E}[\|g^k - \nabla f(x^k)\|^2] \leq (1 - \alpha)\|\nabla f(x^k)\|^2,$$

we have

$$
\begin{aligned}
\mathbb{E}[f(x^{k+1})] &\leq \mathbb{E}[f(x^k)] - \frac{\gamma}{2}\mathbb{E}[\|\nabla f(x^k)\|^2] - (\frac{1}{2\gamma} - \frac{L}{2})\mathbb{E}[\|x^{k+1} - x^k\|^2] + \frac{\gamma}{2}\mathbb{E}[\|g^k - \nabla f(x^k)\|^2] \\
&\leq \mathbb{E}[f(x^k)] - \frac{\gamma}{2}\mathbb{E}[\|\nabla f(x^k)\|^2] - (\frac{1}{2\gamma} - \frac{L}{2})\mathbb{E}[\|x^{k+1} - x^k\|^2] + \frac{\gamma}{2}(1 - \alpha)\mathbb{E}[\|\nabla f(x^k)\|^2] \\
&= \mathbb{E}[f(x^k)] - \frac{\gamma\alpha}{2}\mathbb{E}[\|\nabla f(x^k)\|^2] - (\frac{1}{2\gamma} - \frac{L}{2})\mathbb{E}[\|x^{k+1} - x^k\|^2]
\end{aligned}
$$

If $\gamma \leq 1/L$, we immediately have

$$\frac{\gamma\alpha}{2}\mathbb{E}[\|\nabla f(x^k)\|^2] \leq \mathbb{E}[f(x^k)] - \mathbb{E}[f(x^{k+1})]$$

After telescoping, we get the final result.

$\square$

## B. Details of Convergence Analysis

### B.1. Proof of Lemma 5.7

It is important to note that points within $\text{St}(p, n)^\epsilon$ have bounded singular values. The proof is straightforward and can be referred to (Ablin et al., 2024).

**Lemma B.1** (Singular Values). *If $X \in \text{St}(p, n)^\epsilon$, we have $\sqrt{1-\epsilon} \leq \sigma \leq \sqrt{1+\epsilon}$, for any singular value $\sigma$ of $X$.*

And the proof of Lemma 5.7 is also based on (Ablin et al., 2024).

*Proof.* (Ablin et al., 2024) proved that for any given $X \in \text{St}(p, n)^\epsilon$ and $g \in \mathbb{R}^{n \times p}$, there exists an upper bound of step size $\gamma^*(X; g)$. If step size $\gamma(X; g) \leq \gamma^*(X; g)$, the next iteration remains in $\text{St}(p, n)^\epsilon$. Further, if $\|\text{skew}(gX^\top)X\|_F = a$, the upper-bound of step size $\gamma^*(X; g)$ will not disappear but can be lower-bounded as

$$\gamma^*(X; g) \geq \min\left\{\frac{\lambda(1-\epsilon)\epsilon}{a^2 + \lambda^2(1+\epsilon)\epsilon^2}, \sqrt{\frac{\epsilon}{2a^2}}, \frac{1}{2\lambda}\right\}.$$

Let's take one step further, if $\|\boldsymbol{g}\|_F \leq L'$, we have:

$$\|\text{skew}(\boldsymbol{g}X^\top)X\|_F^2 \leq (1+\epsilon)\|\text{skew}(\boldsymbol{g}X^\top)\|_F^2 \leq (1+\epsilon)\|\boldsymbol{g}X^\top\|_F^2 \leq (1+\epsilon)^2\|\boldsymbol{g}\|_F^2 \leq (1+\epsilon)^2(L')^2, \quad (16)$$

where the first and the third equations are from Lemma B.1 and the second is due to the orthogonality between symmetric and skew-symmetric matrices. Therefore, if we choose a uniform step size

$$\gamma \leq \gamma_s := \min\left\{\frac{\lambda(1-\epsilon)\epsilon}{(1+\epsilon)^2(L')^2 + \lambda^2(1+\epsilon)\epsilon^2}, \sqrt{\frac{\epsilon}{2(1+\epsilon)^2(L')^2}}, \frac{1}{2\lambda}\right\},$$

then for any given $X, \boldsymbol{g}$, the next iteration will remain in the safe region.

$\square$

### B.2. Proof of Lemma 5.8

**Lemma B.2** (Gradient of merit function). *$m(X)$ is the merit function defined in (11), and its gradient can be expressed as*

$$\nabla m(X) = \nabla f(X) - \frac{1}{2}\mathcal{J}_X(\Phi)^*[X^\top X - I_p] - X\text{sym}(X^\top \nabla f(X)) + \mu\nabla\mathcal{N}(X), \quad (17)$$

*where $\Phi(X) = \text{sym}(\nabla f(X)^\top X) : \mathbb{R}^{n \times p} \to \mathbb{R}^{n \times p}$, $\mathcal{J}_X(\Phi)$ denotes its derivative at $X$, and $\mathcal{J}_X(\Phi)^*[X^\top X - I_p]$ denotes the adjoint of the Jacobian in the sense of the Frobenius inner product of $\Phi(X) = \text{sym}(X^\top \nabla f(X))$ in $X$ evaluated in the direction $X^\top X - I_p$. Further, let $\text{vec}(\cdot) : \mathbb{R}^{m \times n} \to \mathbb{R}^{mn}$ denote the vectorization operation, and let $H_X \in \mathbb{R}^{np \times np}$ denote the matrix representation of the Hessian of $f$ at $X$, we have*

$$\text{vec}(\mathcal{J}_X(\Phi)^*[X^\top X - I_p]) = H_X\text{vec}(\nabla\mathcal{N}(X)) + \text{vec}(\nabla f(X)(X^\top X - I_p)). \quad (18)$$

The main part of proof for this lemma can be found at (Ablin et al., 2024).

Now we are ready to prove lemma 5.8.

*Proof.* **STEP 1.** We will bound all the terms from the inner product $\langle\tilde{\Lambda}(X;\boldsymbol{g}), \nabla m(X)\rangle$ separately by considering the four terms in (17) respectively.

The inner product between the first term of (17) and the descent direction is

$$\begin{aligned}
\langle\tilde{\Lambda}(X;\boldsymbol{g}), \nabla f(X)\rangle &= \langle\text{skew}(\boldsymbol{g}X^\top)X + \lambda X(X^\top X - I_p), \nabla f(X)\rangle \\
&= \langle\text{skew}(\boldsymbol{g}X^\top), \nabla f(X)X^\top\rangle + \lambda\langle X^\top X - I_p, X^\top\nabla f(X)\rangle \\
&= \langle\text{skew}(\boldsymbol{g}X^\top), \text{skew}(\nabla f(X)X^\top)\rangle + \lambda\langle X^\top X - I_p, \text{sym}(X^\top\nabla f(X))\rangle. \quad (19)
\end{aligned}$$

The first inner product term can be lower bounded as

$$\begin{aligned}
&\langle\text{skew}(\boldsymbol{g}X^\top), \text{skew}(\nabla f(X)X^\top)\rangle \\
&= \frac{1}{2}\|\text{skew}(\boldsymbol{g}X^\top)\|_F^2 + \frac{1}{2}\|\text{skew}(\nabla f(X)X^\top)\|_F^2 - \frac{1}{2}\|\text{skew}((\boldsymbol{g} - \nabla f(X))X^\top)\|_F^2 \\
&\geq \frac{1}{2}\|\text{skew}(\boldsymbol{g}X^\top)\|_F^2 + \frac{1}{2}\|\text{skew}(\nabla f(X)X^\top)\|_F^2 - \frac{\sigma_1^2}{2}\|\boldsymbol{g} - \nabla f(X)\|_F^2, \quad (20)
\end{aligned}$$

where the last inequality is due to the orthogonality between symmetric and skew-symmetric parts of one matrix, and $\sigma_1$ is the maximum singular value of $X$.

For the inner product between the second term of (17) and the descent direction,

$$\begin{aligned}
\langle\tilde{\Lambda}(X;\boldsymbol{g}), -\frac{1}{2}\mathcal{J}_X(\Phi)^*[X^\top X - I_p]\rangle &= -\frac{1}{2}\lambda\langle H_X\text{vec}(\nabla\mathcal{N}(X)), \text{vec}(\nabla\mathcal{N}(X))\rangle && (21) \\
&\quad -\frac{1}{2}\langle H_X\text{vec}(\nabla\mathcal{N}(X)), \text{vec}(\text{skew}(\boldsymbol{g}X^\top)X)\rangle && (22) \\
&\quad -\frac{1}{2}\langle\nabla f(X)^\top\text{skew}(\boldsymbol{g}X^\top)X, X^\top X - I_p\rangle && (23) \\
&\quad -\frac{1}{2}\lambda\langle\text{sym}(X^\top\nabla f(X)), (X^\top X - I_p)^2\rangle. && (24)
\end{aligned}$$

In the third term of the inner product, the lossy gradient estimate $g$ has no effect:

$$\langle \tilde{\Lambda}(X; g), -X \mathrm{sym}(X^\top \nabla f(X)) \rangle = \langle \mathrm{skew}(gX^\top)X + \lambda \nabla \mathcal{N}(X), -X \mathrm{sym}(X^\top \nabla f(X)) \rangle \tag{25}$$

$$= -\lambda \langle \mathrm{sym}(X^\top \nabla f(X)), X^\top X(X^\top X - I_p) \rangle. \tag{26}$$

The last term is also not affected by the lossy gradient estimate:

$$\langle \tilde{\Lambda}(X; g), \mu \nabla \mathcal{N}(X) \rangle = \lambda \mu \|\nabla \mathcal{N}(X)\|_F^2. \tag{27}$$

Adding all the four terms and applying the lower bound in the first term gives

$$\langle \tilde{\Lambda}(X; g), \nabla m(X) \rangle \geq \frac{1}{2}\|\mathrm{skew}(gX^\top)\|_F^2 + \frac{1}{2}\|\mathrm{skew}(\nabla f(X)X^\top)\|_F^2 - \frac{\sigma_1^2}{2}\|g - \nabla f(X)\|_F^2 \tag{28}$$

$$+ \lambda \langle (\mu I_{np} - \frac{1}{2}H_X)\mathrm{vec}(\nabla \mathcal{N}(X)), \mathrm{vec}(\nabla \mathcal{N}(X)) \rangle \tag{29}$$

$$- \frac{3}{2}\lambda \langle (X^\top X - I_p)^2, \mathrm{sym}(X^\top \nabla f(X)) \rangle \tag{30}$$

$$- \frac{1}{2}\langle H_X \mathrm{vec}(\nabla \mathcal{N}(X)), \mathrm{vec}(\mathrm{skew}(gX^\top)X) \rangle \tag{31}$$

$$- \frac{1}{2}\langle \nabla f(X)^\top \mathrm{skew}(gX^\top)X, X^\top X - I_p \rangle, \tag{32}$$

where the first line (28) comes from the lower bound (20); the second line (29) is a combination of (21) and (27); the third line (30) comes from the second term of (19), (24) and (26); the fourth and the fifth terms are the rest terms (22) and (23).

**STEP 2.** we should separately bound the new five terms. Setting aside the first term, we first analyze the second term (29):

$$\lambda \langle (\mu I_{np} - \frac{1}{2}H_X)\mathrm{vec}(\nabla \mathcal{N}(X)), \mathrm{vec}(\nabla \mathcal{N}(X)) \rangle \geq \lambda(\mu - \frac{L}{2})\|\nabla \mathcal{N}(X)\|_F^2$$

$$\geq 4\lambda(\mu - \frac{L}{2})\sigma_p^2 \mathcal{N}(X), \tag{33}$$

where $L$ is the Lipschitz constant of $\nabla f(X)$ over $\mathrm{St}(p, n)^\epsilon$ defined by Assumption 5.3, and $\sigma_p$ is the minimum singular value of $X$; the first inequality comes from the smoothness of $f(X)$ and the second by the property of singular values of $X$.

The third term (30) can be lower bounded using Cauchy-Schwarz inequality as

$$- \frac{3}{2}\lambda \langle (X^\top X - I_p)^2, \mathrm{sym}(X^\top \nabla f(X)) \rangle \geq -6\lambda \mathcal{N}(X)\|\mathrm{sym}(X^\top \nabla f(X))\|_F$$

$$\geq -6\lambda \sigma_1 L' \mathcal{N}(X), \tag{34}$$

where the second inequality is again due to the orthogonality between symmetric and skew-symmetric matrices and $L'$ is such that $\|\nabla f(X)\|_F \leq L'$ for all $X \in \mathrm{St}(p, n)^\epsilon$ defined by 5.1.

We further use Cauchy-Schwarz inequality to bound the fourth and the fifth terms. The fourth term (31) is lower bounded as

$$- \frac{1}{2}\langle H_X \mathrm{vec}(\nabla \mathcal{N}(X)), \mathrm{vec}(\mathrm{skew}(gX^\top)X) \rangle \geq -\frac{L}{2}\|X(X^\top X - I_p)\|_F\|\mathrm{skew}(gX^\top)X\|_F$$

$$\geq -L\sigma_1 \sqrt{\mathcal{N}(X)}\|\mathrm{skew}(gX^\top)X\|_F. \tag{35}$$

The fifth term is lower bounded as

$$- \frac{1}{2}\langle \nabla f(X)^\top \mathrm{skew}(gX^\top)X, X^\top X - I_p \rangle \geq -\frac{1}{2}\|\nabla f(X)\|_F\|\mathrm{skew}(gX^\top)X\|_F\|X^\top X - I_p\|_F$$

$$\geq -L'\sqrt{\mathcal{N}(X)}\|\mathrm{skew}(gX^\top)X\|_F. \tag{36}$$

Putting the fourth and the fifth terms together, we have

$$
\begin{aligned}
&-\frac{1}{2}\langle H_X \mathrm{vec}(\nabla\mathcal{N}(X)), \mathrm{vec}(\mathrm{skew}(\boldsymbol{g}X^\top)X)\rangle - \frac{1}{2}\langle \nabla f(X)^\top \mathrm{skew}(\boldsymbol{g}X^\top)X, X^\top X - I_p\rangle \\
&\geq -(L' + L\sigma_1)\sqrt{\mathcal{N}(X)}\|\mathrm{skew}(gX^\top)X\|_F \\
&\geq -\frac{1}{2}(L' + L\sigma_1)(b\mathcal{N}(X) + b^{-1}\|\mathrm{skew}(\boldsymbol{g}X^\top)X\|_F^2),
\end{aligned}
\tag{37}
$$

where in the last inequality we use the average inequality $\sqrt{xy} \leq \frac{1}{2}(x+y)$ with $x = b\mathcal{N}(X)$ and $y = b^{-1}\|\mathrm{skew}(\boldsymbol{g}X^\top)X\|_F^2$ for an arbitrary $b > 0$ which will be specified later.

Now for the first term (28), by using the property of singular values of $X$ again, we have

$$
\|\mathrm{skew}(\boldsymbol{g}X^\top)\|_F \geq \sigma_1^{-1}\|\mathrm{skew}(\boldsymbol{g}X^\top)X\|_F, \quad \|\mathrm{skew}(\nabla f(X)X^\top)\|_F \geq \sigma_1^{-1}\|\mathrm{grad}f(X)\|_F.
\tag{38}
$$

Adding all lower bounds (33), (34), (37) and (38) together, we have a total lower bound expressed as

$$
\begin{aligned}
\langle \tilde{\Lambda}(X; \boldsymbol{g}), \nabla m(X)\rangle \geq{}& \frac{1}{2\sigma_1^2}\|\mathrm{grad}f(X)\|_F^2 + \|\mathrm{skew}(\boldsymbol{g}X^\top)X\|_F^2\Big(\frac{1}{2\sigma_1^2} - \frac{1}{2}\frac{(L' + L\sigma_1)}{b}\Big) \\
&- \frac{\sigma_1^2}{2}\|\boldsymbol{g} - \nabla f(X)\|_F^2 + \mathcal{N}(X)\Big(4\lambda(\mu - \frac{L}{2})\sigma_p^2 - 6\lambda\sigma_1 L' - \frac{1}{2}(L' + L\sigma_1)b\Big).
\end{aligned}
\tag{39}
$$

Finally, we choose $b$ so that the coefficient of $\|\mathrm{skew}(\boldsymbol{g}X^\top)X\|_F^2$ equals to $1/4$, namely $b = \frac{2\sigma_1^2(L' + L\sigma_1)}{2 - \sigma_1^2}$, and noticing $\sqrt{1-\epsilon} \leq \sigma_p \leq \sigma_1 \leq \sqrt{1+\epsilon} \leq \sqrt{2}$, we have

$$
\begin{aligned}
\langle \tilde{\Lambda}(X; \boldsymbol{g}), \nabla m(X)\rangle \geq{}& \frac{1}{4}\|\mathrm{grad}f(X)\|_F^2 + \frac{1}{4}\|\mathrm{skew}(\boldsymbol{g}X^\top)X\|_F^2 - \|\boldsymbol{g} - \nabla f(X)\|_F^2 \\
&+ \mathcal{N}(X)\Big(4\lambda(\mu - \frac{L}{2})\sigma_p^2 - 6\lambda\sigma_1 L' - \sigma_1^2\frac{(L' + L\sigma_1)^2}{2 - \sigma_1^2}\Big).
\end{aligned}
\tag{40}
$$

Denoting $\hat{L} = \max\{L', L\}$, the last term of (40) can be simplified as

$$
\begin{aligned}
&\mathcal{N}(X)(4\lambda(\mu - \frac{L}{2})\sigma_p^2 - 6\lambda\sigma_1 L' - \sigma_1^2\frac{(L' + L\sigma_1)^2}{2 - \sigma_1^2} \\
&\geq \mathcal{N}(X)(4\lambda(\mu - \frac{L}{2})(1 - \epsilon) - 6\lambda\sqrt{1+\epsilon}L' - \hat{L}^2(1+\epsilon)\frac{(2\sqrt{1+\epsilon})^2}{2 - (1-\epsilon)}) \\
&\geq \mathcal{N}(X)(4\lambda(\mu - \frac{L}{2})(1 - \epsilon) - 6\lambda\sqrt{1+\epsilon}L' - 4\hat{L}^2(1+\epsilon)) \\
&\geq \lambda\mu\mathcal{N}(X),
\end{aligned}
\tag{41}
$$

where in the last inequality, we choose $\mu$ as

$$
\mu \geq \frac{2}{3 - 4\epsilon}\Big(L(1 - \epsilon) + 3\sqrt{1+\epsilon}L' + 2\hat{L}^2\frac{1+\epsilon}{\lambda}\Big),
\tag{42}
$$

which leads to the conclusion. $\qquad\square$

### B.3. Proof of Theorem 5.9

We introduce some notation for convenience.

- Auxiliary variables: momentum vector of the master $v^k := \frac{1}{N}\sum_{i=1}^N v_i^k$; random variable of the master $\boldsymbol{\xi}^k := (\xi_1^k, \ldots, \xi_N^k)$; stochastic gradient of the master $\nabla F(X^k; \boldsymbol{\xi}^k) := \frac{1}{N}\sum_{i=1}^N \nabla F(X^k; \xi_i^k)$.

- Momentum and stochastic error: error of each node $P_i^k := \|v_i^k - \nabla f_i(X^k)\|_F^2$; averaged error $\tilde{P}^k := \frac{1}{N}\sum_{i=1}^N \|v_i^k - \nabla f_i(X^k)\|_F^2$; error of the master $P^k := \|v^k - \nabla f(X^k)\|_F^2$.

- Compression error: error of each node $G_i^k := \|\boldsymbol{g}_i^k - \boldsymbol{v}_i^k\|_F^2$; averaged error $\tilde{G}^k := \frac{1}{N}\sum_{i=1}^N \|\boldsymbol{g}_i^k - \boldsymbol{v}_i^k\|_F^2$; error of the master $G^k := \|\boldsymbol{g}^k - \boldsymbol{v}^k\|_F^2$.

- Filtrations: filtration for the conditional expectation of stochastic gradient $\mathcal{F}_P^k := \{\boldsymbol{\xi}^0, X^1, \boldsymbol{\xi}^1, X^2, \dots, \boldsymbol{\xi}^{k-1}, X^k\}$; filtration for the conditional expectation of compressor $\mathcal{F}_C^k := \{\boldsymbol{\xi}^0, X^1, \boldsymbol{\xi}^1, X^2, \dots, \boldsymbol{\xi}^{k-1}, X^k, \boldsymbol{\xi}^k\}$.

Next, for arbitrary momentum factor $\eta$, The following two lemmas provide the iterative formats of two kinds of errors.

**Lemma B.3.** *The iteration of momentum and stochastic error satisfies*

$$\mathbb{E}[P_i^{k+1}] \le (1-\eta)\mathbb{E}[P_i^k] + (1-\eta)^2(1+\frac{1}{\eta})L_i^2\mathbb{E}\big[\|X^{k+1} - X^k\|_F^2\big] + \eta^2\sigma^2; \tag{43}$$

$$\mathbb{E}[\tilde{P}^{k+1}] \le (1-\eta)\mathbb{E}[\tilde{P}^k] + (1-\eta)^2(1+\frac{1}{\eta})\tilde{L}^2\mathbb{E}\big[\|X^{k+1} - X^k\|_F^2\big] + \eta^2\sigma^2; \tag{44}$$

$$\mathbb{E}[P^{k+1}] \le (1-\eta)\mathbb{E}[P^k] + (1-\eta)^2(1+\frac{1}{\eta})L^2\mathbb{E}\big[\|X^{k+1} - X^k\|_F^2\big] + \frac{\eta^2\sigma^2}{N}. \tag{45}$$

*Proof.* For the first inequality (43),

$$
\begin{aligned}
\mathbb{E}[P_i^{k+1}] &= \mathbb{E}\big[\|\boldsymbol{v}_i^{k+1} - \nabla f_i(X^{k+1})\|_F^2\big] \\
&= \mathbb{E}\big[\|(1-\eta)\boldsymbol{v}_i^k + \eta\nabla F(X^{k+1}; \xi_i^{k+1}) - \nabla f_i(X^{k+1})\|_F^2\big] \\
&= \mathbb{E}\Big[\big\|(1-\eta)\big(\boldsymbol{v}_i^k - \nabla f_i(X^{k+1})\big) + \eta\big(\nabla F(X^{k+1}; \xi_i^{k+1}) - \nabla f_i(X^{k+1})\big)\big\|_F^2\Big] \\
&= \mathbb{E}\Big[\mathbb{E}_{\xi_i^{k+1}}\Big[\big\|(1-\eta)\big(\boldsymbol{v}_i^k - \nabla f_i(X^{k+1})\big) + \eta\big(\nabla F(X^{k+1}; \xi_i^{k+1}) - \nabla f_i(X^{k+1})\big)\big\|_F^2\Big|\mathcal{F}_P^{k+1}\Big]\Big] \\
&\le \mathbb{E}\big[(1-\eta)^2\|\boldsymbol{v}_i^k - \nabla f_i(X^{k+1})\|_F^2\big] + \eta^2\sigma^2 \\
&\le (1-\eta)^2(1+\eta)\mathbb{E}\big[\|\boldsymbol{v}_i^k - \nabla f_i(X^k)\|_F^2\big] + (1-\eta)^2(1+\frac{1}{\eta})\mathbb{E}\big[\|\nabla f_i(X^{k+1}) - \nabla f_i(X^k)\|_F^2\big] + \eta^2\sigma^2 \\
&\le (1-\eta)\mathbb{E}[P_i^k] + (1-\eta)^2(1+\frac{1}{\eta})L_i^2\mathbb{E}\big[\|X^{k+1} - X^k\|_F^2\big] + \eta^2\sigma^2,
\end{aligned}
\tag{46}
$$

where the second equation is by the update rule of $\boldsymbol{v}_i^{k+1}$, the first inequality is due to Assumption 5.6, the second inequality is the consequence of Young's inequality and the last is by the smoothness of each local function.

By simply summing all inequalities in terms of node from 1 to $N$, we get the second inequality (44). Noticing that $\nabla F(X^k; \boldsymbol{\xi}^k) := \frac{1}{N}\sum_{i=1}^N \nabla F(X^k; \xi_i^k)$ and according to the independency of each entry of $\boldsymbol{\xi}$, we have $\mathbb{E}_{\xi^{k+1}}\big[\|\nabla F(X^{k+1}; \boldsymbol{\xi}^{k+1}) - \nabla f(X^{k+1})\|_F^2\big|\mathcal{F}_P^{k+1}\big] \le \sigma^2/N$. Similarly, we can derive the third inequality (45).

$\square$

**Lemma B.4.** *The iteration of compression error satisfies*

$$\mathbb{E}[G_i^{k+1}] \le (1-\theta)\mathbb{E}[G_i^k] + 2\beta\eta^2 L_i^2\mathbb{E}\big[\|X^{k+1} - X^k\|_F^2\big] + 2\beta\eta^2\mathbb{E}[P_i^k] + (1-\alpha)\eta^2\sigma^2; \tag{47}$$

$$\mathbb{E}[\tilde{G}^{k+1}] \le (1-\theta)\mathbb{E}[\tilde{G}^k] + 2\beta\eta^2\tilde{L}^2\mathbb{E}\big[\|X^{k+1} - X^k\|_F^2\big] + 2\beta\eta^2\mathbb{E}[\tilde{P}^k] + (1-\alpha)\eta^2\sigma^2, \tag{48}$$

*where $\theta$ and $\beta$ are two scalars related to the contractive factor $\alpha$, i.e., $\theta := 1 - \sqrt{1-\alpha}$ and $\beta := \frac{1-\alpha}{1-\sqrt{1-\alpha}}$.*

*Proof.* For the first inequality (47),

$$
\begin{aligned}
\mathbb{E}[G_i^{k+1}] = \mathbb{E}\big[\|\boldsymbol{g}_i^{k+1} - \boldsymbol{v}_i^{k+1}\|_F^2\big] &= \mathbb{E}\big[\|\boldsymbol{g}_i^k + \boldsymbol{c}_i^k - \boldsymbol{v}_i^{k+1}\|_F^2\big] \\
&= \mathbb{E}\big[\|(\boldsymbol{v}_i^{k+1} - \boldsymbol{g}_i^k) - \mathcal{C}(\boldsymbol{v}_i^{k+1} - \boldsymbol{g}_i^k)\|_F^2\big] = \mathbb{E}\Big[\mathbb{E}_\mathcal{C}\big[\|(\boldsymbol{v}_i^{k+1} - \boldsymbol{g}_i^k) - \mathcal{C}(\boldsymbol{v}_i^{k+1} - \boldsymbol{g}_i^k)\|_F^2\big|\mathcal{F}_\mathcal{C}^{k+1}\big]\Big] \\
&\leq (1-\alpha)\mathbb{E}\big[\|\boldsymbol{v}_i^{k+1} - \boldsymbol{g}_i^k\|_F^2\big] = (1-\alpha)\mathbb{E}\big[\|(1-\eta)\boldsymbol{v}_i^k + \eta\nabla F(X^{k+1};\xi_i^{k+1}) - \boldsymbol{g}_i^k\|_F^2\big] \\
&= (1-\alpha)\mathbb{E}\Big[\big\|\boldsymbol{v}_i^k - \boldsymbol{g}_i^k + \eta\big(\nabla f_i(X^{k+1}) - \boldsymbol{v}_i^k\big) + \eta\big(\nabla F(X^{k+1};\xi_i^{k+1}) - \nabla f(X^{k+1})\big)\big\|_F^2\Big] \\
&= (1-\alpha)\mathbb{E}\Big[\mathbb{E}_{\xi_i^{k+1}}\big[\big\|\boldsymbol{v}_i^k - \boldsymbol{g}_i^k + \eta\big(\nabla f_i(X^{k+1}) - \boldsymbol{v}_i^k\big) + \eta\big(\nabla F(X^{k+1};\xi_i^{k+1}) - \nabla f(X^{k+1})\big)\big\|_F^2\big|\mathcal{F}_P^{k+1}\big]\Big] \\
&\leq (1-\alpha)\mathbb{E}\Big[\big\|\boldsymbol{v}_i^k - \boldsymbol{g}_i^k + \eta\big(\nabla f_i(X^{k+1}) - \boldsymbol{v}_i^k\big)\big\|_F^2\Big] + (1-\alpha)\eta^2\sigma^2 \\
&\leq (1-\alpha)(1+\rho)\mathbb{E}\big[\|\boldsymbol{g}_i^k - \boldsymbol{v}_i^k\|_F^2\big] + (1-\alpha)(1+\frac{1}{\rho})\eta^2\mathbb{E}\big[\|\nabla f_i(X^{k+1}) - \boldsymbol{v}_i^k\|_F^2\big] + (1-\alpha)\eta^2\sigma^2, \quad (49)
\end{aligned}
$$

where the first inequality is due the property of contractive compressors (Definition 3.2), the second inequality is by the bounded variance (Assumption 5.6) and the third is the consequence of Young's inequality with parameter $\rho$. We should choose a proper $\rho$ satisfying $(1-\alpha)(1+\rho) < 1$ and $(1-\alpha)(1+1/\rho) < +\infty$. (Richtarik et al., 2021) provided a to some degree optimal choice $\rho = \frac{1}{\sqrt{1-\alpha}} - 1$. Denoting $1 - \theta = (1-\alpha)(1+\rho) = \sqrt{1-\alpha}$, and $\beta = (1-\alpha)(1+1/\rho) = \frac{1-\alpha}{1-\sqrt{1-\alpha}}$, we have

$$
\begin{aligned}
\mathbb{E}[G_i^{k+1}] &\leq (1-\theta)\mathbb{E}\big[\|\boldsymbol{g}_i^k - \boldsymbol{v}_i^k\|_F^2\big] + \beta\eta^2\mathbb{E}\big[\|\nabla f_i(X^{k+1}) - \boldsymbol{v}_i^k\|_F^2\big] + (1-\alpha)\eta^2\sigma^2 \\
&\leq (1-\theta)\mathbb{E}\big[\|\boldsymbol{g}_i^k - \boldsymbol{v}_i^k\|_F^2\big] + 2\beta\eta^2\mathbb{E}\big[\|\nabla f_i(X^{k+1}) - \nabla f_i(X^k)\|_F^2\big] \\
&\quad + 2\beta\eta^2\mathbb{E}\big[\|\nabla f_i(X^k) - \boldsymbol{v}_i^k\|_F^2\big] + (1-\alpha)\eta^2\sigma^2 \\
&\leq (1-\theta)\mathbb{E}[G_i^k] + 2\beta\eta^2 L_i^2\mathbb{E}\big[\|X^{k+1} - X^k\|_F^2\big] + 2\beta\eta^2\mathbb{E}[P_i^k] + (1-\alpha)\eta^2\sigma^2. \quad (50)
\end{aligned}
$$

The second inequality (48) follows the same process as lemma B.3. Lacking the unbiasedness assumption of the compressor, we have no similar inequality for master, which is fortunately unnecessary. □

Another lemma helps us split $\|X^{k+1} - X^k\|_F^2$ term to two orthogonal component used for convergence analysis.

**Lemma B.5.**

$$
\|X^{k+1} - X^k\|_F^2 = \gamma^2\|\tilde{\Lambda}(X^k;\tilde{\boldsymbol{g}}^k)\|_F^2 \leq \gamma^2\Big(\big\|\mathrm{skew}\big(\tilde{\boldsymbol{g}}^k(X^k)^\top\big)\big\|_F^2 + 4\lambda^2(1+\epsilon)\mathcal{N}(X^k)\Big) \quad (51)
$$

*Proof.* This lemma can be easily proved by the orthogonality between two terms of $\tilde{\Lambda}(X^k;\tilde{\boldsymbol{g}}^k)$ and the sigular values bound of $X^k$ (Lemma B.1). □

Now we are able to prove the main theorem.

*Proof.* **STEP 1. Dividing different Errors.** Using smoothness of $m(X^k)$ mentioned in Section 3.1 and Lemma 5.8, we have:

$$
\begin{aligned}
m(X^{k+1}) &\leq m(X^k) + \langle\nabla m(X^k), X^{k+1} - X^k\rangle + \frac{L_m}{2}\|X^{k+1} - X^k\|_F^2 \\
&= m(X^k) - \gamma\langle\nabla m(X^k), \tilde{\Lambda}(X^k;\tilde{\boldsymbol{g}}^k)\rangle + \frac{L_m\gamma^2}{2}\|\tilde{\Lambda}(X^k;\tilde{\boldsymbol{g}}^k)\|_F^2 \\
&\leq m(X^k) - \frac{\gamma}{4}\|\mathrm{grad}f(X^k)\|_F^2 - \frac{\gamma}{4}\big\|\mathrm{skew}\big(\tilde{\boldsymbol{g}}^k(X^k)^\top\big)X^k\big\|_F^2 + \gamma\|\tilde{\boldsymbol{g}}^k - \nabla f(X^k)\|_F^2 \\
&\quad - \gamma\lambda\mu\mathcal{N}(X^k) + \frac{L_m\gamma^2}{2}\|\tilde{\Lambda}(X^k;\tilde{\boldsymbol{g}}^k)\|_F^2. \quad (52)
\end{aligned}
$$

Denote $\mathbb{S} := \{X \in \mathbb{R}^{n \times p} | \|X\|_F \leq L\}$, then we have $\tilde{\boldsymbol{g}}^k = \operatorname{proj}_{\mathbb{S}}(\boldsymbol{g}^k)$, and $\nabla f(X^k) = \operatorname{proj}_{\mathbb{S}}(\nabla f(X^k))$. Therefore, using the property of projections,

$$\|\tilde{\boldsymbol{g}}^k - \nabla f(X^k)\|_F^2 \leq \|\boldsymbol{g}^k - \nabla f(X^k)\|_F^2 \tag{53}$$

Next, we need to separete the momentum and stochastic error as well as compression error. We first introduce an inequality $\|\boldsymbol{g}^k - \nabla f(X^k)\|_F^2 \leq c_1 \|\boldsymbol{g}^k - \boldsymbol{v}^k\|_F^2 + c_2 \|\boldsymbol{v}^k - \nabla f(X^k)\|_F^2$, where two undetermined constants are specified in three cases:

- When the momentum rate $\eta = 1$ and the variance of gradients $\sigma^2 = 0$, the momentum and stochastic error will be 0, so $c_1, c_2$ are specified as $c_1 = 1, c_2 = 0$, which is trivial.

- When the contractive factor $\alpha = 1$, the compression error will be 0, so $c_1, c_2$ are specified as $c_1 = 0, c_2 = 1$, which is trivial too.

- Otherwise, we specify $c_1 = c_2 = 2$, which is the consequence of mean value inequality.

Therefore,

$$\begin{aligned}
\|\boldsymbol{g}^k - \nabla f(X^k)\|_F^2 &\leq c_1 \|\boldsymbol{g}^k - \boldsymbol{v}^k\|_F^2 + c_2 \|\boldsymbol{v}^k - \nabla f(X^k)\|_F^2 \\
&\leq \frac{c_1}{N} \sum_{i=1}^N \|\boldsymbol{g}_i^k - \boldsymbol{v}_i^k\|_F^2 + c_2 \|\boldsymbol{v}^k - \nabla f(X^k)\|_F^2 = c_1 \tilde{G}^k + c_2 P^k,
\end{aligned} \tag{54}$$

where the second inequality is due to Jensen's inequality.

For convenience, we denote $S^k = \|\operatorname{skew}(\tilde{\boldsymbol{g}}^k (X^k)^\top) X^k\|_F^2$, whose influence will be eliminated later by choosing a proper step size. Substituting (54) into (52) after using (53) , we have:

$$m(X^{k+1}) \leq m(X^k) - \frac{\gamma}{4} \|\operatorname{grad} f(X^k)\|_F^2 - \frac{\gamma}{4} S^k + \gamma c_1 \tilde{G}^k + \gamma c_2 P^k - \gamma \lambda \mu \mathcal{N}(X^k) + \frac{L_m \gamma^2}{2} \|\tilde{\Lambda}(X^k; \tilde{\boldsymbol{g}}^k)\|_F^2 \tag{55}$$

**STEP 2. Constructing Lyapunov Function.** Adding $\frac{c_1 \gamma}{\theta} G^{k+1}$ to both sides of 55, taking expectation and using lemma B.4, we have:

$$\begin{aligned}
&\mathbb{E}[m(X^{k+1})] + \frac{c_1 \gamma}{\theta} \mathbb{E}[\tilde{G}^{k+1}] \\
&\leq \mathbb{E}[m(X^k)] - \frac{\gamma}{4} \mathbb{E}[\|\operatorname{grad} f(X^k)\|_F^2] - \frac{\gamma}{4} \mathbb{E}[S^k] \\
&\quad + \frac{c_1 \gamma}{\theta} \left( \mathbb{E}[\tilde{G}^k] + 2\beta \eta^2 \tilde{L}^2 \mathbb{E}[\|X^{k+1} - X^k\|_F^2] + 2\beta \eta^2 \mathbb{E}[\tilde{P}^k] + (1-\alpha)\eta^2 \sigma^2 \right) \\
&\quad + c_2 \gamma \mathbb{E}[P^k] - \gamma \lambda \mu \mathbb{E}[\mathcal{N}(X^k)] + \frac{L_m \gamma^2}{2} \mathbb{E}[\|\tilde{\Lambda}(X^k; \tilde{\boldsymbol{g}}^k)\|_F^2] \\
&= \mathbb{E}[m(X^k)] + \frac{c_1 \gamma}{\theta} \mathbb{E}[\tilde{G}^k] \\
&\quad - \frac{\gamma}{4} \mathbb{E}[\|\operatorname{grad} f(X^k)\|_F^2] - \frac{\gamma}{4} \mathbb{E}[S^k] - \gamma \lambda \mu \mathbb{E}[\mathcal{N}(X^k)] \\
&\quad + \left( \frac{2c_1 \gamma^3 \eta^2 \beta \tilde{L}^2}{\theta} + \frac{L_m \gamma^2}{2} \right) \mathbb{E}[\|\tilde{\Lambda}(X^k; \tilde{\boldsymbol{g}}^k)\|_F^2] \\
&\quad + \frac{2c_1 \gamma \eta^2 \beta}{\theta} \mathbb{E}[\tilde{P}^k] + c_2 \gamma \mathbb{E}[P^k] + \frac{c_1 \gamma \eta^2 (1-\alpha)\sigma^2}{\theta}.
\end{aligned} \tag{56}$$

Further adding $\frac{2c_1 \gamma \eta \beta}{\theta} \tilde{P}^{k+1}$, $\frac{c_2 \gamma}{\eta} P^{k+1}$ to both sides, subtracting lower bound $m^*$ by Assumption 5.5, and using lemma B.3,

we have the following collated inequality

$$\mathbb{E}[m(X^{k+1})] - m^* + \frac{c_1\gamma}{\theta}\mathbb{E}[\tilde{G}^{k+1}] + \frac{2c_1\gamma\eta\beta}{\theta}\mathbb{E}[\tilde{P}^{k+1}] + \frac{c_2\gamma}{\eta}\mathbb{E}[P^{k+1}]$$

$$\leq \mathbb{E}[m(X^k)] - m^* + \frac{c_1\gamma}{\theta}\mathbb{E}[\tilde{G}^k] + \frac{2c_1\gamma\eta\beta}{\theta}\mathbb{E}[\tilde{P}^k] + \frac{c_2\gamma}{\eta}\mathbb{E}[P^k]$$

$$- \frac{\gamma}{4}\mathbb{E}[\|\mathrm{grad}f(X^k)\|_F^2] - \frac{\gamma}{4}\mathbb{E}[S^k] - \gamma\lambda\mu\mathbb{E}[\mathcal{N}(X^k)]$$

$$+ \Big(\frac{2c_1\beta\tilde{L}^2}{\theta}(1-\eta+\eta^3)\gamma^3 + \frac{c_2L^2}{\eta^2}(1-\eta)^2(1+\eta)\gamma^3 + \frac{L_m\gamma^2}{2}\Big)\mathbb{E}[\|\tilde{\Lambda}(X^k;\tilde{\boldsymbol{g}}^k)\|_F^2]$$

$$+ \frac{c_1\gamma\eta^2(1-\alpha)\sigma^2}{\theta} + \frac{2c_1\gamma\eta^3\beta\sigma^2}{\theta} + \frac{c_2\gamma\eta\sigma^2}{N}, \tag{57}$$

which corresponds to the Lyapunov function we defined as (12).

**STEP 3. Choosing Step Size.** Next, we split $\|\tilde{\Lambda}(X^k;\tilde{\boldsymbol{g}}^k)\|_F^2$ using lemma B.5, which splits $\mathbb{E}[\|\tilde{\Lambda}(X^k;\tilde{\boldsymbol{g}}^k)\|_F^2]$ into $\mathbb{E}[S^k]$ and $\mathbb{E}[\mathcal{N}(X^k)]$. To eliminate the influence of $S^k$ and to make the coefficient before $\mathcal{N}(X^k)$ negative enough, we are looking for a sufficiently small step size to make the coefficients before $\mathbb{E}[S^k]$ and $\mathbb{E}[\mathcal{N}(X^k)]$ satisfy

$$-\frac{\gamma}{4} + \frac{2c_1\beta\tilde{L}^2}{\theta}(1-\eta+\eta^3)\gamma^3 + \frac{c_2L^2}{\eta^2}(1-\eta)^2(1+\eta)\gamma^3 + \frac{L_m\gamma^2}{2} \leq 0 \tag{58}$$

and

$$-\gamma\lambda\mu + 4\lambda^2(1+\epsilon)\Big(\frac{2c_1\beta\tilde{L}^2}{\theta}(1-\eta+\eta^3)\gamma^3 + \frac{c_2L^2}{\eta^2}(1-\eta)^2(1+\eta)\gamma^3 + \frac{L_m\gamma^2}{2}\Big) \leq -\frac{\gamma\lambda\mu}{2}. \tag{59}$$

It's reducible to a quadratic inequality problem. suppose $a, b > 0$, if we choose $0 \leq x \leq (\sqrt{a} + b)^{-1}$, the inequality $ax^2 + bx \leq 1$ holds. So it's enough to choose step size $\gamma$ as

$$\gamma = \min\{\gamma_s, \gamma_1, \gamma_2\},$$

where $\gamma_s$ is the uniform safe step size defined by Lemma 5.7, and $\gamma_1, \gamma_2$ are defined as

$$\gamma_1 := \big(2\sqrt{a} + 2L_m\big)^{-1}$$

$$\gamma_2 := \Big(2\sqrt{\frac{2\lambda^2(1+\epsilon)a}{\mu}} + \frac{4\lambda^2(1+\epsilon)L_m}{\mu}\Big)^{-1},$$

where

$$a = \frac{2c_1\beta\tilde{L}^2}{\theta}(1-\eta+\eta^3) + \frac{c_2L^2}{\eta^2}(1-\eta)^2(1+\eta).$$

With a small enough step size $\gamma$, we have

$$\mathbb{E}[\mathcal{L}^{k+1}] \leq \mathbb{E}[\mathcal{L}^k] - \frac{\gamma}{4}\mathbb{E}[\|\mathrm{grad}f(X^k)\|_F^2] - \frac{\gamma\lambda\mu}{2}\mathbb{E}[\mathcal{N}(X^k)] + \frac{c_1\gamma\eta^2(1-\alpha)\sigma^2}{\theta} + \frac{2c_1\gamma\eta^3\beta\sigma^2}{\theta} + \frac{c_2\gamma\eta\sigma^2}{N} \tag{60}$$

**STEP 4. Summing Up.** Summing up all inequalities from $k = 0$ to $K - 1$ and dividing by $K$, we have

$$\frac{1}{4K}\sum_{k=0}^{K-1}\mathbb{E}[\|\mathrm{grad}f(X^k)\|_F^2] + \frac{\lambda\mu}{2K}\sum_{k=0}^{K-1}\mathbb{E}[\mathcal{N}(X^k)] \leq \frac{\mathcal{L}^0}{\gamma K} + \frac{c_1\eta^2(1-\alpha)\sigma^2}{\theta} + \frac{2c_1\eta^3\beta\sigma^2}{\theta} + \frac{c_2\eta\sigma^2}{N}, \tag{61}$$

$\square$

## B.4. Details of Theorem 5.11 and Corollary 5.12

**1. Theorem 5.11.** When $\sigma = 0$, $\eta = 1$, $c_1 = 1$ and $c_2 = 0$, the momentum and stochastic error $\tilde{P}^k = 0$, so the Lyapunov function defined in (12) can be simplified as

$$\mathcal{L}^k = m(X^k) - m^* + \frac{\gamma}{\theta}\tilde{G}^k. \tag{62}$$

According to the rule of choosing step size in Appendix B.3, the step size should be chosen as

$$\gamma = \min\{\gamma_s, \gamma_1, \gamma_2\},$$

where $\gamma_s$ is the uniform safe step size defined by Lemma 5.7, and $\gamma_1, \gamma_2$ are defined as

$$\gamma_1 := \left(2\sqrt{a} + 2L_m\right)^{-1}$$

$$\gamma_2 := \left(2\sqrt{\frac{2\lambda^2(1+\epsilon)a}{\mu}} + \frac{4\lambda^2(1+\epsilon)L_m}{\mu}\right)^{-1},$$

where

$$a = \frac{2\beta\tilde{L}^2}{\theta}.$$

**2. Theorem 5.12.** When $\alpha = 1$, the compression error $\tilde{G}^k$ will be 0, so the Lyapunov function can be further simplified as $\mathcal{L}^k = m(X^k) - m^*$. And the choice of the step size can also be simplified as

$$\gamma = \min\left\{\gamma_s, \frac{1}{2L_m}, \frac{\mu}{4\lambda^2(1+\epsilon)L_m}\right\}$$

.

## B.5. An adequate Selection of Step Size with momentum

Before deriving conclusions about convergence with momentum, we give a lemma about the step size. This lemma gives an adequate but not necessary choice for step size, for the convenience of analyzing the influence of momentum.

**Lemma B.6** (An adequate Selection of Step Size with momentum). *For the step size defined in Theorem 5.9, fix momentum rate $\eta \in (0,1)$. If we further choose $\gamma = \min\{\gamma_s, \frac{1}{6\tilde{L}}\sqrt{\frac{\theta}{2c_1\beta}}, \frac{\eta}{6L\sqrt{2c_2}}, \frac{1}{6L_m}, \frac{1}{12\tilde{L}}\sqrt{\frac{\mu\theta}{c_1\lambda^2(1+\epsilon)\beta}}, \frac{\eta}{12L}\sqrt{\frac{\mu}{c_2\lambda^2(1+\epsilon)}}, \frac{\mu}{12\lambda^2(1+\epsilon)L_m}\}$, then the step size satisfies $\gamma \le \min\{\gamma_s, \gamma_1, \gamma_2\}$ defined in Theorem 5.9.*

*Proof.* The rule of choosing step size in Appendix B.3 is

$$\gamma = \min\{\gamma_s, \gamma_1, \gamma_2\},$$

where $\gamma_s$ is the uniform safe step size defined by Lemma 5.7, and $\gamma_1, \gamma_2$ are defined as

$$\gamma_1 := \left(2\sqrt{a} + 2L_m\right)^{-1}$$

$$\gamma_2 := \left(2\sqrt{\frac{2\lambda^2(1+\epsilon)a}{\mu}} + \frac{4\lambda^2(1+\epsilon)L_m}{\mu}\right)^{-1},$$

where

$$a = \frac{2c_1\beta\tilde{L}^2}{\theta}(1 - \eta + \eta^3) + \frac{c_2L^2}{\eta^2}(1-\eta)^2(1+\eta).$$

We can bound $\sqrt{a}$ as

$$\sqrt{a} = \sqrt{\frac{2c_1\beta\tilde{L}^2}{\theta}(1-\eta+\eta^3) + \frac{c_2L^2}{\eta^2}(1-\eta)^2(1+\eta)} \le \sqrt{\frac{2c_1\beta\tilde{L}^2}{\theta} + \frac{2c_2L^2}{\eta^2}} \le \tilde{L}\sqrt{\frac{2c_1\beta}{\theta}} + \frac{L}{\eta}\sqrt{2c_2}, \tag{63}$$

where the last inequality is due to $\sqrt{x+y} \leq \sqrt{x} + \sqrt{y}$, $x, y \geq 0$.

Therefore,

$$\gamma_1 = (2\sqrt{a} + 2L_m)^{-1} \geq \left(2\tilde{L}\sqrt{\frac{2c_1\beta}{\theta}} + 2\frac{L}{\eta}\sqrt{2c_2} + 2L_m\right)^{-1}. \tag{64}$$

For $(x+y+z)^{-1}$, $x, y, z > 0$, if we denote $M = \max\{x, y, z\}$, then $\frac{1}{M} = \min\{\frac{1}{x}, \frac{1}{y}, \frac{1}{z}\}$. So $(x+y+z)^{-1} \geq \frac{1}{3M} = \frac{1}{3}\min\{\frac{1}{x}, \frac{1}{y}, \frac{1}{z}\}$. Hence, if we choose $\gamma \leq \min\{\frac{1}{6\tilde{L}}\sqrt{\frac{\theta}{2c_1\beta}}, \frac{\eta}{6L\sqrt{2c_2}}, \frac{1}{6L_m}\}$, then it satisfies $\gamma \leq \gamma_1$.

The same reasoning leads to

$$\gamma_2 = \left(2\sqrt{\frac{2\lambda^2(1+\epsilon)a}{\mu}} + \frac{4\lambda^2(1+\epsilon)L_m}{\mu}\right)^{-1} \geq \left(4\tilde{L}\sqrt{\frac{c_1\lambda^2(1+\epsilon)\beta}{\mu\theta}} + 4\frac{L}{\eta}\sqrt{\frac{c_2\lambda^2(1+\epsilon)}{\mu}} + \frac{4\lambda^2(1+\epsilon)L_m}{\mu}\right)^{-1}. \tag{65}$$

Choosing $\gamma \leq \min\{\frac{1}{12\tilde{L}}\sqrt{\frac{\mu\theta}{c_1\lambda^2(1+\epsilon)\beta}}, \frac{\eta}{12L}\sqrt{\frac{\mu}{c_2\lambda^2(1+\epsilon)}}, \frac{\mu}{12\lambda^2(1+\epsilon)L_m}\}$ satisfies $\gamma \leq \gamma_2$. And the conclusion follows.

$\square$

### B.6. Proof of Theorem 5.14

*Proof.* When $c_1 = c_2 = 2$, the Lyapunov function is

$$\mathcal{L}^k = m(X^k) - m^* + \frac{2\gamma}{\theta}\tilde{G}^k + \frac{4\gamma\eta\beta}{\theta}\tilde{P}^k + \frac{2\gamma}{\eta}P^k. \tag{66}$$

Choosing step size $\gamma$ as Lemma B.6 and setting $c_1 = c_2 = 2$ leads to

$$\frac{1}{\gamma} \leq \left(\frac{1}{\gamma_s} + 6L_m + \frac{12\lambda^2(1+\epsilon)L_m}{\mu} + 12\tilde{L}\sqrt{\frac{\beta}{\theta}} + 12\tilde{L}\sqrt{\frac{2\lambda^2(1+\epsilon)\beta}{\mu\theta}}\right) + \left(12L + 12L\sqrt{\frac{2\lambda^2(1+\epsilon)}{\mu}}\right) \cdot \frac{1}{\eta}. \tag{67}$$

We denote $C_1^\gamma = \frac{1}{\gamma_s} + 6L_m + \frac{12\lambda^2(1+\epsilon)L_m}{\mu} + 12\tilde{L}\sqrt{\frac{\beta}{\theta}} + 12\tilde{L}\sqrt{\frac{2\lambda^2(1+\epsilon)\beta}{\mu\theta}}$ and $C_2^\gamma = 12L + 12L\sqrt{\frac{2\lambda^2(1+\epsilon)}{\mu}}$ for convenience. Using Theorem 5.9, we have

$$\frac{1}{4K}\sum_{k=0}^{K-1}\mathbb{E}[\|\text{grad}f(X^k)\|_F^2] + \frac{\lambda\mu}{2K}\sum_{k=0}^{K-1}\mathbb{E}[\mathcal{N}(X^k)] \leq \frac{C_1^\gamma\mathcal{L}^0}{K} + \frac{C_2^\gamma\mathcal{L}^0}{\eta K} + \frac{2\eta^2(1-\alpha)\sigma^2}{\theta} + \frac{4\eta^3\beta\sigma^2}{\theta} + \frac{2\eta\sigma^2}{N}. \tag{68}$$

We choose $\eta = \min\{(\frac{C_2^\gamma N\mathcal{L}^0}{2\sigma^2 K})^{1/2}, (\frac{\theta C_2^\gamma\mathcal{L}^0}{2(1-\alpha)\sigma^2 K})^{1/3}, (\frac{\theta C_2^\gamma\mathcal{L}^0}{4\beta\sigma^2 K})^{1/4}\}$, so that $\frac{C_2^\gamma\mathcal{L}^0}{\eta K} \geq \frac{2\eta\sigma^2}{N}$, $\frac{C_2^\gamma\mathcal{L}^0}{\eta K} \geq \frac{2\eta^2(1-\alpha)\sigma^2}{\theta}$ and $\frac{C_2^\gamma\mathcal{L}^0}{\eta K} \geq \frac{4\eta^3\beta\sigma^2}{\theta}$, and at least one inequality takes equal. Hence, (68) can be bounded as

$$\frac{1}{4K}\sum_{k=0}^{K-1}\mathbb{E}[\|\text{grad}f(X^k)\|_F^2] + \frac{\lambda\mu}{2K}\sum_{k=0}^{K-1}\mathbb{E}[\mathcal{N}(X^k)] \leq \frac{C_1^\gamma\mathcal{L}^0}{K} + 4\frac{C_2^\gamma\mathcal{L}^0}{\eta K}$$
$$\leq \frac{C_1^\gamma\mathcal{L}^0}{K} + 4\left(\frac{2\eta\sigma^2}{N} + \frac{2\eta^2(1-\alpha)\sigma^2}{\theta} + \frac{4\eta^3\beta\sigma^2}{\theta}\right)$$
$$\leq \frac{C_1^\gamma\mathcal{L}^0}{K} + 4\left(\frac{2\sigma^2 C_2^\gamma\mathcal{L}^0}{NK}\right)^{\frac{1}{2}} + 4\left(\frac{2\sigma^2(1-\alpha)}{\theta}\right)^{\frac{1}{3}}\left(\frac{C_2^\gamma\mathcal{L}^0}{K}\right)^{\frac{2}{3}} + 4\left(\frac{4\beta\sigma^2}{\theta}\right)^{\frac{1}{4}}\left(\frac{C_2^\gamma\mathcal{L}^0}{K}\right)^{\frac{3}{4}}, \tag{69}$$

which leads to the conclusion. $\square$

## B.7. Proof of Theorem 5.16

*Proof.* When $c_1 = 0, c_2 = 1$, the Lyapunov function can be simplified as

$$\mathcal{L}^k = m(X^k) - m^* + \frac{\gamma}{\eta} P^k. \tag{70}$$

And Theorem 5.9 leads to

$$\frac{1}{4K} \sum_{k=0}^{K-1} \mathbb{E}[\|\text{grad}f(X^k)\|_F^2] + \frac{\lambda\mu}{2K} \sum_{k=0}^{K-1} \mathbb{E}[\mathcal{N}(X^k)] \leq \frac{\mathcal{L}^0}{\gamma K} + \frac{\eta\sigma^2}{N}. \tag{71}$$

Choosing step size $\gamma$ as Lemma B.6 and setting $c_1 = 0, c_2 = 1$ leads to

$$\frac{1}{\gamma} \leq \left( \frac{1}{\gamma_s} + 6L_m + \frac{12\lambda^2(1+\epsilon)L_m}{\mu} \right) + \left( 6\sqrt{2}L + 12L\sqrt{\frac{\lambda^2(1+\epsilon)}{\mu}} \right) \cdot \frac{1}{\eta}. \tag{72}$$

We denote $C_1^\gamma = \frac{1}{\gamma_s} + 6L_m + \frac{12\lambda^2(1+\epsilon)L_m}{\mu}$ and $C_2^\gamma = 6\sqrt{2}L + 12L\sqrt{\frac{\lambda^2(1+\epsilon)}{\mu}}$ for convenience, and (71) leads to

$$\frac{1}{4K} \sum_{k=0}^{K-1} \mathbb{E}[\|\text{grad}f(X^k)\|_F^2] + \frac{\lambda\mu}{2K} \sum_{k=0}^{K-1} \mathbb{E}[\mathcal{N}(X^k)] \leq \frac{C_1^\gamma \mathcal{L}^0}{K} + \frac{C_2^\gamma \mathcal{L}^0}{\eta K} + \frac{\eta\sigma^2}{N}. \tag{73}$$

Choosing $\eta = \sqrt{\frac{C_2^\gamma N \mathcal{L}^0}{\sigma^2 K}}$ leads to the conclusion.

$\square$

# C. Details of Optimization on Block-wise Manifolds

In this section, we provide a block-wise version of EF-Landing algorithm. And we further prove the convergence result for stochastic scenarios with compression. Other Scenarios can be deduced similarly.

## C.1. Algorithm for Block-wise Problems

---

**Algorithm 3** Block-wise EF-Landing

---

**Require:** starting point $\mathfrak{X}^0 \in \mathfrak{R}$; gradient bound $L'$; step size $\gamma > 0$; compressor $\mathcal{C}$; momentum $\eta \in (0, 1]$;

1: Each node initializes $\mathfrak{v}_i^0 = \nabla F(\mathfrak{X}^0; \xi_i^0)$, $\mathfrak{g}_i^0 = \mathcal{C}(\mathfrak{v}_i^0)$ for $i = 1, \ldots, N$; Master initializes $\mathfrak{g}^0 = \frac{1}{N} \sum_{i=1}^N \mathfrak{g}_i^0$.

2: **for** $k = 0, 1, \ldots, K - 1$ **do**

3:      Master clips the gradient $\tilde{\mathfrak{g}}^k = \min\{1, L'/\|\mathfrak{g}^k\|\}\mathfrak{g}^k$;

4:      Master computes $\tilde{\Lambda}(\mathfrak{X}^k; \tilde{\mathfrak{g}}^k)$ using (14);

5:      Master computes $\mathfrak{X}^{k+1} = \mathfrak{X}^k - \gamma\tilde{\Lambda}(\mathfrak{X}^k; \tilde{\mathfrak{g}}^k)$ and broadcasts $\mathfrak{X}^{k+1}$ to all nodes.

6:      **for** all nodes $i = 1, \ldots, N$ in parallel **do**

7:          Compute momentum $\mathfrak{v}_i^{k+1} = (1 - \eta)\mathfrak{v}_k^k + \eta\nabla F(\mathfrak{X}^{k+1}; \xi_i^{k+1})$;

8:          Compress $\mathfrak{c}_i^k$ via $\mathfrak{c}_i^k = \mathcal{C}(\mathfrak{v}_i^{k+1} - \mathfrak{g}_i^k)$ and send it to the master;

9:          Update local state $\mathfrak{g}_i^{k+1}$ via $\mathfrak{g}_i^{k+1} = \mathfrak{g}_i^k + \mathfrak{c}_i^k$.

10:      **end for**

11:      Master updates $\mathfrak{g}^{k+1}$ via $\mathfrak{g}^{k+1} = \mathfrak{g}^k + \frac{1}{N} \sum_{i=1}^N \mathfrak{c}_i^k$.

12: **end for**

---

## C.2. Convergence Results of EF-Landing in Stochastic Scenarios for Block-wise Constraints

**Theorem C.1** (Block-wise EF-Landing Convergence in Stochastic Scenarios). *Consider the problem* (13) *with block-wise constraints* (13). *Letting Assumptions* 5.1, 5.3, 5.5 *and* 5.6 *hold, if compressor* $\mathcal{C}$ *satisfies Definition* 3.2 *(all in the sense of composite data type in* $\mathfrak{R}$ *), if we choose step size* $\gamma$ *as in Lemma* B.6, *by running Algorithm* 1 *for* $K$ *iterations, we have*

$$\frac{1}{K}\sum_{k=0}^{K-1}\sum_{j=1}^{J}\mathbb{E}[\|\mathrm{grad}_j f(X_j^k)\|_F^2] \leq 4\mathcal{O}\Big(\frac{1}{\sqrt{NK}}\Big),$$

$$\frac{1}{K}\sum_{k=0}^{K-1}\sum_{j=1}^{J}\mathbb{E}[\mathcal{N}_j(X_j^k)] \leq \frac{2}{\lambda\mu}\mathcal{O}\Big(\frac{1}{\sqrt{NK}}\Big),$$

$$\frac{1}{K}\sum_{k=0}^{K-1}\mathbb{E}\Big[\Big\|\frac{\partial f}{\partial x^k}\Big\|_2^2\Big] \leq 2\mathcal{O}\Big(\frac{1}{\sqrt{NK}}\Big),$$

*where* $\mathcal{O}(\frac{1}{\sqrt{NK}})$ *is specified as*

$$\mathcal{O}\Big(\frac{1}{\sqrt{NK}}\Big) = \frac{C_1^\gamma \mathcal{L}^0}{K} + 4\Big(\frac{2\sigma^2 C_2^\gamma \mathcal{L}^0}{NK}\Big)^{\frac{1}{2}}$$
$$+ 4\Big(\frac{2\sigma^2(1-\alpha)}{\theta}\Big)^{\frac{1}{3}}\Big(\frac{C_2^\gamma \mathcal{L}^0}{K}\Big)^{\frac{2}{3}} + 4\Big(\frac{4\beta\sigma^2}{\theta}\Big)^{\frac{1}{4}}\Big(\frac{C_2^\gamma \mathcal{L}^0}{K}\Big)^{\frac{3}{4}},$$

*when choosing* $\eta = \min\{(\frac{C_2^\gamma N\mathcal{L}^0}{2\sigma^2 K})^{1/2}, (\frac{\theta C_2^\gamma \mathcal{L}^0}{2(1-\alpha)\sigma^2 K})^{1/3}, (\frac{\theta C_2^\gamma \mathcal{L}^0}{4\beta\sigma^2 K})^{1/4}\}$. $\mathcal{L}^0$ *is defined in* (12), $\theta$ *and* $\beta$ *are defined in Theorem* 5.9, $C_1^\gamma, C_2^\gamma$ *are two constants defined as* $C_1^\gamma = \frac{1}{\gamma_s} + 6L_m + \frac{12\lambda^2(1+\epsilon)L_m}{\mu} + 12\tilde{L}\sqrt{\frac{\beta}{\theta}} + 12\tilde{L}\sqrt{\frac{2\lambda^2(1+\epsilon)\beta}{\mu\theta}}$ *and* $C_2^\gamma = 12L + 12L\sqrt{\frac{2\lambda^2(1+\epsilon)}{\mu}}$.

*Proof.* We choose the merit function as:

$$m(\mathfrak{X}) = f(\mathfrak{X}) - \sum_{j=1}^{J}h_j(X_j) + \sum_{j=1}^{J}\mu\mathcal{N}_j(X_j), \tag{74}$$

where $h_j(X_j) = \frac{1}{2}\Big\langle \mathrm{sym}\Big(X_j^\top \frac{\partial f}{\partial X_j}\Big), X_j^\top X_j - I_{p_j} \Big\rangle$ and $\mu$ is a hyper-parameter specified later.

Noticing that partial functions like $h_j(X_j)$ satisfies $\frac{\partial h_j}{\partial X_k} = 0, \forall j \neq k$, we have

$$\langle\tilde{\Lambda}(\mathfrak{X};\mathfrak{g}), \nabla m(\mathfrak{X})\rangle = \sum_{j=1}^{J}\Big\langle \mathrm{skew}(\boldsymbol{g}_j X_j^\top)X_j + \lambda X_j(X_j^\top X_j - I_p), \frac{\partial f}{\partial X_j} - \nabla h_j(X_j) + \nabla\mu\mathcal{N}_j(X_j)\Big\rangle + \Big\langle \boldsymbol{g}_0, \frac{\partial f}{\partial x}\Big\rangle.$$

Noticing that for all $j = 1, \ldots, J$, $\frac{\partial f}{\partial X_j}$ is $L$-Lipschitz smooth as long as $\nabla f(\mathfrak{X})$ is $L$-Lipschitz smooth, and $\frac{\partial f}{\partial X_j}$ is $L'$-bounded as long as $\nabla f(\mathfrak{X})$ is $L'$-bounded, using Lemma 5.8 with the same choice of $\mu$, we have

$$\langle\tilde{\Lambda}(\mathfrak{X};\mathfrak{g}), \nabla m(\mathfrak{X})\rangle \geq \frac{1}{4}\sum_{j=1}^{J}\Big\|\mathrm{skew}\Big(\frac{\partial f}{\partial X_j}X_j^\top\Big)X_j\Big\|_F^2 + \frac{1}{4}\sum_{j=1}^{J}\Big\|\mathrm{skew}\Big(\boldsymbol{g}_j X_j^\top\Big)X_j^\top\Big\|_F^2 - \sum_{j=1}^{J}\Big\|\boldsymbol{g}_j - \frac{\partial f}{\partial X_j}\Big\|_F^2$$
$$+ \sum_{j=1}^{J}\lambda\mu\mathcal{N}(X_j) + \frac{1}{2}\|\boldsymbol{g}_0\|_2^2 + \frac{1}{2}\Big\|\frac{\partial f}{\partial x}\Big\|_2^2 - \Big\|\boldsymbol{g}_0 - \frac{\partial f}{\partial x}\Big\|_2^2, \tag{75}$$

where for $\langle\boldsymbol{g}_0, \frac{\partial f}{\partial x}\rangle$ we use $\langle a, b\rangle = \frac{1}{2}\|a\|_2^2 + \frac{1}{2}\|b\|_2^2 - \frac{1}{2}\|a-b\|_2^2 \geq \frac{1}{2}\|a\|_2^2 + \frac{1}{2}\|b\|_2^2 - \|a-b\|_2^2$. The inequality is for the sake of $\|\mathfrak{g} - \nabla f(\mathfrak{X})\|^2 = \sum_{j=1}^{J}\|\boldsymbol{g}_j - \frac{\partial f}{\partial X_j}\|_F^2 + \|\boldsymbol{g}_0 - \frac{\partial f}{\partial x}\|_2^2$. Hence, using smoothness of $m(\mathfrak{X}^k)$ and the merit function

bound (75), we have

$$
\begin{aligned}
m(\mathfrak{X}^{k+1}) &\leq m(\mathfrak{X}^k) + \langle m(\mathfrak{X}^k), \mathfrak{X}^{k+1} - \mathfrak{X}^k \rangle + \frac{L_m}{2}\|\mathfrak{X}^{k+1} - \mathfrak{X}^k\|^2 \\
&= m(\mathfrak{X}^k) - \gamma\langle\nabla m(\mathfrak{X}^k), \tilde{\Lambda}(\mathfrak{X}^k; \tilde{\mathfrak{g}}^k)\rangle + \frac{L_m\gamma^2}{2}\|\tilde{\Lambda}(\mathfrak{X}^k; \tilde{\mathfrak{g}}^k)\|^2 \\
&\leq m(\mathfrak{X}^k) - \frac{\gamma}{4}\sum_{j=1}^{J}\|\mathrm{grad}_j f(X_j^k)\|_F^2 - \frac{\gamma}{4}\sum_{j=1}^{J}\|\mathrm{skew}(\tilde{\boldsymbol{g}}_j^k(X_j^k)^\top)X_j^k\|_F^2 - \gamma\lambda\mu\sum_{j=1}^{J}\mathcal{N}_j(X_j^k) \\
&\quad + \frac{L_m\gamma^2}{2}\|\tilde{\Lambda}(\mathfrak{X}^k; \tilde{\mathfrak{g}}^k)\|^2 - \frac{\gamma}{2}\|\tilde{\boldsymbol{g}}_0^k\|_2^2 - \frac{\gamma}{2}\left\|\frac{\partial f}{\partial x^k}\right\|_2^2 + \gamma\|\tilde{\mathfrak{g}}^k - \nabla f(\mathfrak{X}^k)\|^2
\end{aligned}
\tag{76}
$$

Denote $S^k = \sum_{j=1}^{J}\|\mathrm{skew}(\tilde{\boldsymbol{g}}_j^k(X_j^k)^\top)X_j^k\|_F^2$, following the same process as the proof of Theorem 5.9, we have

$$
\begin{aligned}
&\mathbb{E}[m(\mathfrak{X}^{k+1})] - m^* + \frac{c_1\gamma}{\theta}\mathbb{E}[\tilde{G}^{k+1}] + \frac{2c_1\gamma\eta\beta}{\theta}\mathbb{E}[\tilde{P}^{k+1}] + \frac{c_2\gamma}{\eta}\mathbb{E}[P^{k+1}] \\
&\leq \mathbb{E}[m(\mathfrak{X}^k)] - m^* + \frac{c_1\gamma}{\theta}\mathbb{E}[\tilde{G}^k] + \frac{2c_1\gamma\eta\beta}{\theta}\mathbb{E}[\tilde{P}^k] + \frac{c_2\gamma}{\eta}\mathbb{E}[P^k] \\
&\quad - \frac{\gamma}{4}\sum_{j=1}^{J}\mathbb{E}[\|\mathrm{grad}_j f(X_j^k)\|_F^2] - \frac{\gamma}{4}\mathbb{E}[S^k] - \gamma\lambda\mu\sum_{j=1}^{J}\mathbb{E}[\mathcal{N}_j(X_j^k)] - \frac{\gamma}{2}\mathbb{E}\left[\left\|\frac{\partial f}{\partial x^k}\right\|_2^2\right] - \frac{\gamma}{2}\mathbb{E}[\|\tilde{\boldsymbol{g}}_0^k\|_2^2] \\
&\quad + \left(\frac{2c_1\beta\tilde{L}^2}{\theta}(1 - \eta + \eta^3)\gamma^3 + \frac{c_2L^2}{\eta^2}(1 - \eta)^2(1 + \eta)\gamma^3 + \frac{L_m\gamma^2}{2}\right)\mathbb{E}[\|\tilde{\Lambda}(\mathfrak{X}^k; \tilde{\mathfrak{g}}^k)\|^2] \\
&\quad + \frac{c_1\gamma\eta^2(1 - \alpha)\sigma^2}{\theta} + \frac{2c_1\gamma\eta^3\beta\sigma^2}{\theta} + \frac{c_2\gamma\eta\sigma^2}{N}.
\end{aligned}
\tag{77}
$$

Noticing that $\|\tilde{\Lambda}(\mathfrak{X}^k; \tilde{\mathfrak{g}}^k)\|^2 \leq \sum_{j=1}^{J}\|\mathrm{skew}(\tilde{\boldsymbol{g}}_j(X_j^k)^\top)X_j^k\|_F^2 + 4\lambda^2(1 + \epsilon)\sum_{j=1}^{J}\mathcal{N}_j(X_j^k) + \|\tilde{\boldsymbol{g}}_0\|_2^2$, if we choose $\gamma \leq \gamma_1$ defined in Theorem 5.9, the following inequality also holds, so that the influence of $\|\tilde{\boldsymbol{g}}_0\|_2^2$ can be eliminated.

$$
-\frac{\gamma}{2} + \frac{2c_1\beta\tilde{L}^2}{\theta}(1 - \eta + \eta^3)\gamma^3 + \frac{c_2L^2}{\eta^2}(1 - \eta)^2(1 + \eta)\gamma^3 + \frac{L_m\gamma^2}{2} \leq 0.
\tag{78}
$$

Hence, we do not need any additional restriction for $\gamma$ apart from $\gamma_s, \gamma_1, \gamma_2$ in Theorem 5.9. With the step size above, we have

$$
\begin{aligned}
\mathbb{E}[\mathcal{L}^{k+1}] &\leq \mathbb{E}[\mathcal{L}^k] - \frac{\gamma}{4}\sum_{j=1}^{J}\mathbb{E}[\|\mathrm{grad}_j f(X_j^k)\|_F^2] - \frac{\gamma\lambda\mu}{2}\sum_{j=1}^{J}\mathbb{E}[\mathcal{N}_j(X_j^k)] + \frac{\gamma}{2}\mathbb{E}\left[\left\|\frac{\partial f}{\partial x^k}\right\|_2^2\right] \\
&\quad + \frac{c_1\gamma\eta^2(1 - \alpha)\sigma^2}{\theta} + \frac{2c_1\gamma\eta^3\beta\sigma^2}{\theta} + \frac{c_2\gamma\eta\sigma^2}{N}.
\end{aligned}
\tag{79}
$$

Summing up all inequalities from $k = 0$ to $K - 1$ and dividing by $K$, we have

$$
\begin{aligned}
&\frac{1}{4K}\sum_{k=0}^{K-1}\sum_{j=1}^{J}\mathbb{E}[\|\mathrm{grad}f(X_j^k)\|_F^2] + \frac{\lambda\mu}{2K}\sum_{k=0}^{K-1}\sum_{j=0}^{J}\mathbb{E}[\mathcal{N}_j(X_j^k)] + \frac{1}{2K}\sum_{k=0}^{K-1}\left\|\frac{\partial f}{\partial x^k}\right\|_2^2 \\
&\leq \frac{\mathcal{L}^0}{\gamma K} + \frac{c_1\eta^2(1 - \alpha)\sigma^2}{\theta} + \frac{2c_1\eta^3\beta\sigma^2}{\theta} + \frac{c_2\eta\sigma^2}{N}.
\end{aligned}
\tag{80}
$$

Follow the same process of proof of Theorem 5.14, we achieve the final result.

$\square$

# D. Further Numerical Experiments

## D.1. Online PCA

### D.1.1. DATASETS

In the experiment, the matrix $A \in \mathbb{R}^{N \times n}$ is synthetically generated, where $N = 20,000$ represents the number of samples, each of dimension $n = 5000$. The columns of $A$ are independently sampled from the normal distribution $\mathcal{N}(0, UU^T + \sigma I_n)$, with $\sigma = 0.1$, and $U \in \mathbb{R}^{n \times p}$ is sampled from the Stiefel manifold using the uniform Haar distribution.

### D.1.2. HYPERPARAMETERS

We conducted the experiment in deterministic scenarios and distributed the data across 4 nodes. The EF-Landing algorithm was tested for different values of $p$ (100, 200, 500, and 1000). The results were compared with the QR retraction method, Euclidean gradient descent with $\ell_2$ regularization (referred to as the Penalty method), and the Landing algorithm. The compression methods used included Top-$K$, Random-$K$ and QSGD. For the Top-$K$ and Random-$K$ compression, the compression rate was set to 0.1, and for QSGD, the quantization level $s$ was set to 8 for $p = 100$ and $p = 200$, and 16 for $p = 500$ and $p = 1000$. The experiment involved 600 iterations for all algorithms with a fixed learning rate, except for the QSGD algorithm when $p = 1000$, where a learning rate decay was applied during the iterations.

| Method/Hyperparameter | $\gamma$ | Clipping | $\lambda$ | compress rate /s |
|---|---|---|---|---|
| EF-Landing-Top-$K$ | 1.0 | $10^8$ | 0.5 | 0.1 |
| EF-Landing-Random-$K$ | 1.0 | $10^8$ | 0.5 | 0.1 |
| EF-Landing-QSGD | 1.0 (0.01 after 100 steps) | $10^8$ | 0.5 | 8 |
| Landing | 1.0 | $10^8$ | 0.5 | – |
| Retraction | 1.0 | – | 0.5 | – |
| Penalty method | 1.0 | – | 8.0 | – |

Table 2: Hyperparameters of $p = 100$

| Method/Hyperparameter | $\gamma$ | Clipping | $\lambda$ | compress rate /s |
|---|---|---|---|---|
| EF-Landing-Top-$K$ | 1.0 | $10^8$ | 0.5 | 0.1 |
| EF-Landing-Random-$K$ | 1.0 | $10^8$ | 0.5 | 0.1 |
| EF-Landing-QSGD | 1.0 (0.01 after 100 steps) | $10^8$ | 0.5 | 8 |
| Landing | 1.0 | $10^8$ | 0.5 | – |
| Retraction | 1.0 | – | 0.5 | – |
| Penalty method | 1.0 | – | 8.0 | – |

Table 3: Hyperparameters of $p = 200$

| Method/Hyperparameter | $\gamma$ | Clipping | $\lambda$ | compress rate /s |
|---|---|---|---|---|
| EF-Landing-Top-$K$ | 1.0 (0.01 after 100 steps) | $10^8$ | 0.5 | 0.1 |
| EF-Landing-Random-$K$ | 1.0 | $10^8$ | 0.5 | 0.1 |
| EF-Landing-QSGD | 1.0 (0.1 after 50 steps) | $10^8$ | 0.5 | 16 |
| Landing | 1.0 | $10^8$ | 0.5 | – |
| Retraction | 1.0 | – | 0.5 | – |
| Penalty method | 1.0 | – | 8.0 | – |

Table 4: Hyperparameters of $p = 500$

| Method/Hyperparameter | $\gamma$ | Clipping | $\lambda$ | compress rate /s |
|---|---|---|---|---|
| EF-Landing-Top-$K$ | 1.0 | $10^8$ | 0.5 | 0.1 |
| EF-Landing-Random-$K$ | 1.0 | $10^8$ | 0.5 | 0.1 |
| EF-Landing-QSGD | 1.0 (0.1 after 100 steps) | $10^8$ | 0.5 | 16 |
| Landing | 1.0 | $10^8$ | 0.5 | – |
| Retraction | 1.0 | – | 0.5 | – |
| Penalty method | 1.0 | – | 8.0 | – |

Table 5: Hyperparameters of $p = 1000$

### D.1.3. EXPERIMENT RESULTS

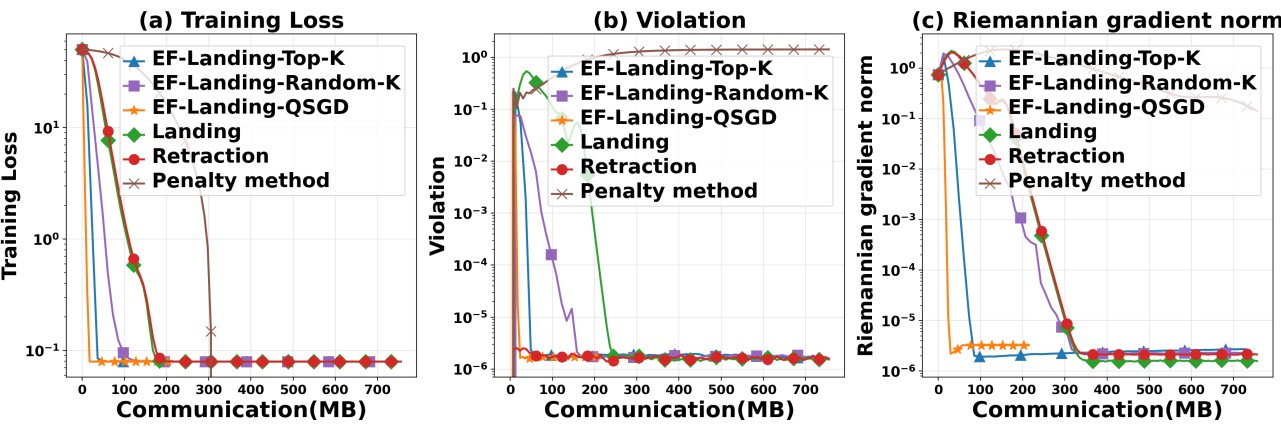

Figure 5: Performance of EF-Landing and other algorithms on online PCA, with $p = 100$

The corresponding experimental results for parameter $p$ taking values of 100, 200, and 500 are shown in Figures 5, 6, and 7, respectively. We can observe that, regardless of the compression algorithm, EF-Landing consistently reduces the communication volume by at least half while still converging to the optimal value of the function and ensuring that the norm of Riemannian gradient approaches 0. Additionally, compared to the Landing algorithm, the EF-Landing algorithm requires less communication to satisfy the constraints on the variable $X$, specifically ensuring that $\mathcal{N}(X)$ is sufficiently small, as demonstrated in Figures 6 and 7.

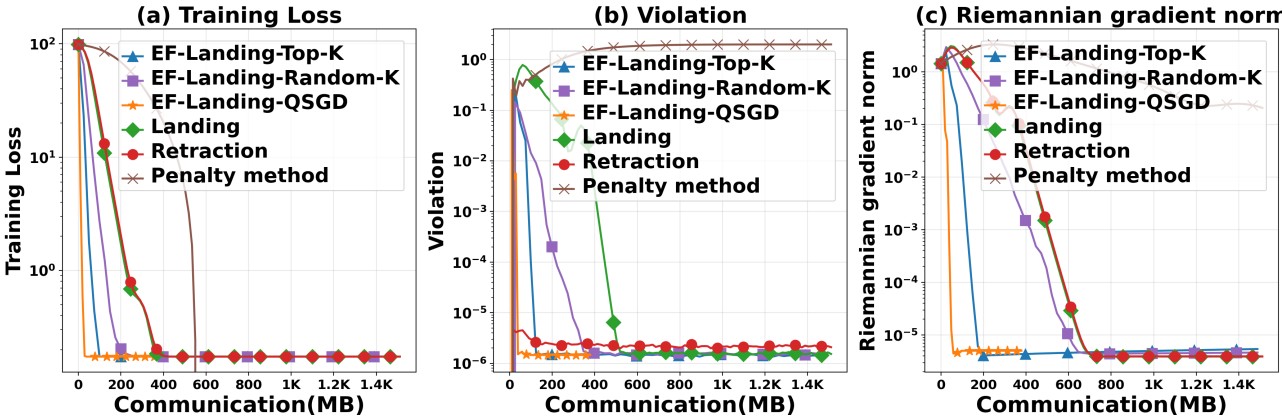

Figure 6: Performance of EF-Landing and other algorithms on online PCA, with $p = 200$

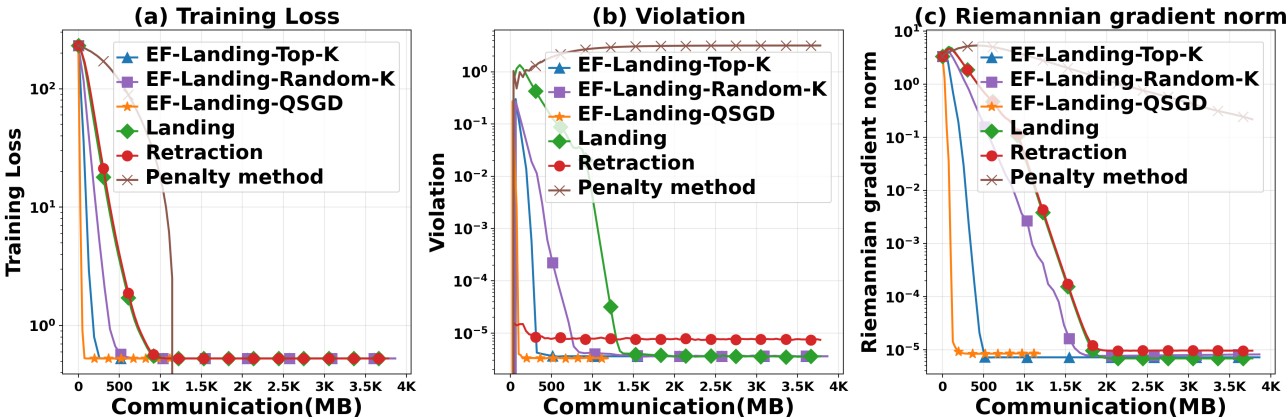

Figure 7: Performance of EF-Landing and other algorithms on online PCA, with $p = 500$

Compared to the QR retraction method, EF-Landing reduces a significant amount of computational effort by eliminating the need for QR decomposition, making the algorithm more efficient. On the other hand, the Penalty method is highly sensitive to the choice of $\lambda$. If $\lambda$ is too small, the parameters may fail to satisfy the constraint, as shown in Figures 2 and 5 to 7. If $\lambda$ is too large, the Riemannian gradient may fail to converge to zero and can lead to numerical instability. Although the Landing method can more accurately satisfy the orthogonality constraints, it requires several times more communication. By contrast, the EF-Landing algorithm remains highly efficient.

### D.2. Neural Networks with Orthogonality Constraints

We also tested the performance of the EF-Landing algorithm on neural network models with orthogonal constraints applied to the convolutional layers. Orthogonal constraints are playing an increasingly important role in deep learning. For example, they can ensure the stability of gradient magnitudes during training (Arjovsky et al., 2016; Saxe et al., 2014), preventing issues like gradient explosion or vanishing. Furthermore, there has been growing attention on how to design orthogonal convolutions (Singla & Feizi, 2021; Boissin et al., 2025; Yu et al., 2022).

#### D.2.1. DATASETS

The MNIST dataset only has a citation requirement (LeCun et al., 2010). It includes $28 \times 28$ grayscale images of handwritten digits from 0 to 9, containing 60,000 training data samples and 10,000 test data samples.

The CIFAR-10 dataset, which consists of $32 \times 32$ color images depicting 10 distinct categories of real-world objects, is comprised of 50,000 training samples and 10,000 testing samples. It requires citation (Krizhevsky et al., 2009) for usage.

We applied some basic data augmentation techniques to these datasets during the training stage. For CIFAR-10, we applied random cropping, random horizontal flipping and random gray scale.

We conducted experiments on VGG16 and ResNet-18, comparing the performance of the EF-Landing algorithm with the Landing and QR retraction algorithms. Specifically, we applied orthogonal constraints to the convolutional layers by reshaping the convolutional kernels to the size $n_{out} \times n_{in} n_x n_y$, where $n_{in}$ and $n_{out}$ represent the number of input and output channels, respectively, and $n_x$ and $n_y$ are the filter dimensions (Ablin et al., 2024). In cases where the reshaped matrix becomes wide instead of tall, we enforce orthogonality on its transpose.It is worth noting that the Block-wise Constraints formulation in (13) better suits this problem, as we impose constraints on each individual convolutional layer rather than on the entire neural network parameters $X$.

#### D.2.2. VGG16 ON MNIST

In this experiment, we use the built-in VGG16 model from PyTorch and initialize the convolutional layers using QR decomposition to enforce orthogonal constraints. To match the model architecture, we added a convolutional layer to the VGG16 model. This convolutional layer has an output channel size of 3 and a padding of 3. Consequently, we adjusted the shape of the data to $32 \times 32$. The compression methods applied are Top-$K$ and Rand-$K$, with a compression ratio of 0.2. The training data is uniformly distributed across 4 nodes. The initial learning rate $\gamma$ is determined via grid search over the

| Method/Hyperparameter | $\gamma$ | $\eta$ | Clipping | $\lambda$ | compress rate |
|---|---|---|---|---|---|
| EF-Landing-Top-$K$ | 0.1 | 0.1 | $10^8$ | 1.0 | 0.2 |
| EF-Landing-Random-$K$ | 0.1 | 0.5 | $10^8$ | 1.0 | 0.2 |
| Landing | 0.1 | – | – | 1.0 | – |
| Retraction | 0.1 | – | – | 1.0 | – |

Table 6: Hyperparameters: VGG16 on MNIST

set $\{0.001, 0.01, 0.1, 1.0\}$, and is decayed by a factor of $1/10$ every 10 epochs. Similarly, the step size $\eta$ is selected from $\{0.1, 0.5, 0.7\}$ via grid search. We use a batch size of 128 and train the model for a total of 30 epochs.

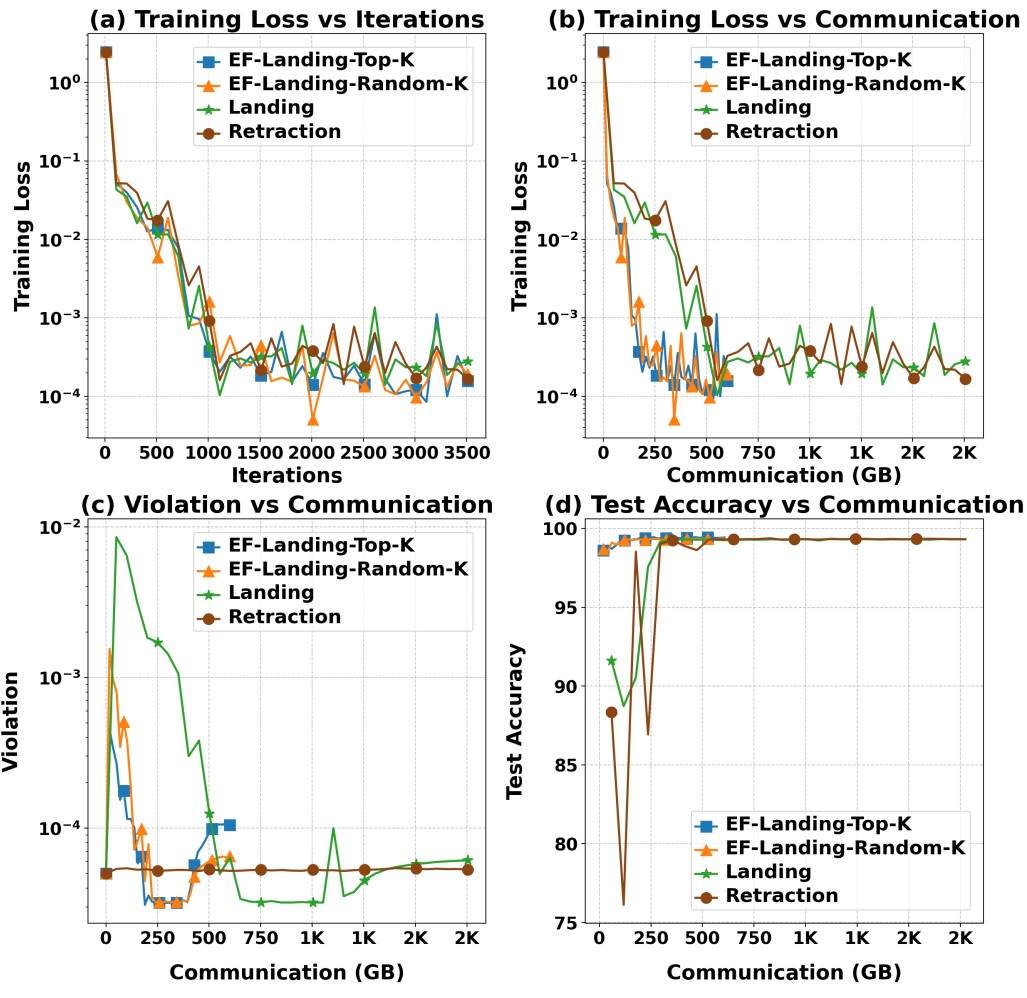

Figure 8: Performance of EF-Landing and other algorithms on deep learning with VGG16 on MNIST.

Figure 8 demonstrates that the EF-Landing algorithm achieves convergence of the training loss function and test accuracy to a steady state with less communication volume, and the accuracy on the test set stabilizes around 99%. In addition, the EF-Landing algorithm outperforms the Landing algorithm in satisfying the orthogonality constraint under this experiment, which also highlights the advantage of the EF-Landing algorithm to some extent.

| Method/Hyperparameter | $\gamma$ | $\eta$ | Clipping | $\lambda$ | compress rate |
|---|---|---|---|---|---|
| EF-Landing-Top-$K$ | 0.1 | 0.5 | $10^8$ | 1.0 | 0.2 |
| EF-Landing-Random-$K$ | 0.1 | 0.1 | $10^8$ | 1.0 | 0.2 |
| Landing | 0.1 | – | – | 1.0 | – |
| Retraction | 0.1 | – | – | 1.0 | – |

Table 7: Hyperparameters: VGG16 on CIFAR-10

### D.2.3. VGG16 AND RESNET-18 ON CIFAR-10

Additionally, we tested the performance of EF-Landing on the CIFAR-10 dataset in a 4-node setting, using the Top-$K$ and Rand-$K$ compressors with a compression ratio of 0.2. We compared the results with the Landing and QR retraction methods. In the experiment, each algorithm was trained for 150 epochs, and after the 100th epoch, the learning rate was reduced to $\frac{1}{10}$ of its original value, with $\lambda$ set to 1.0.

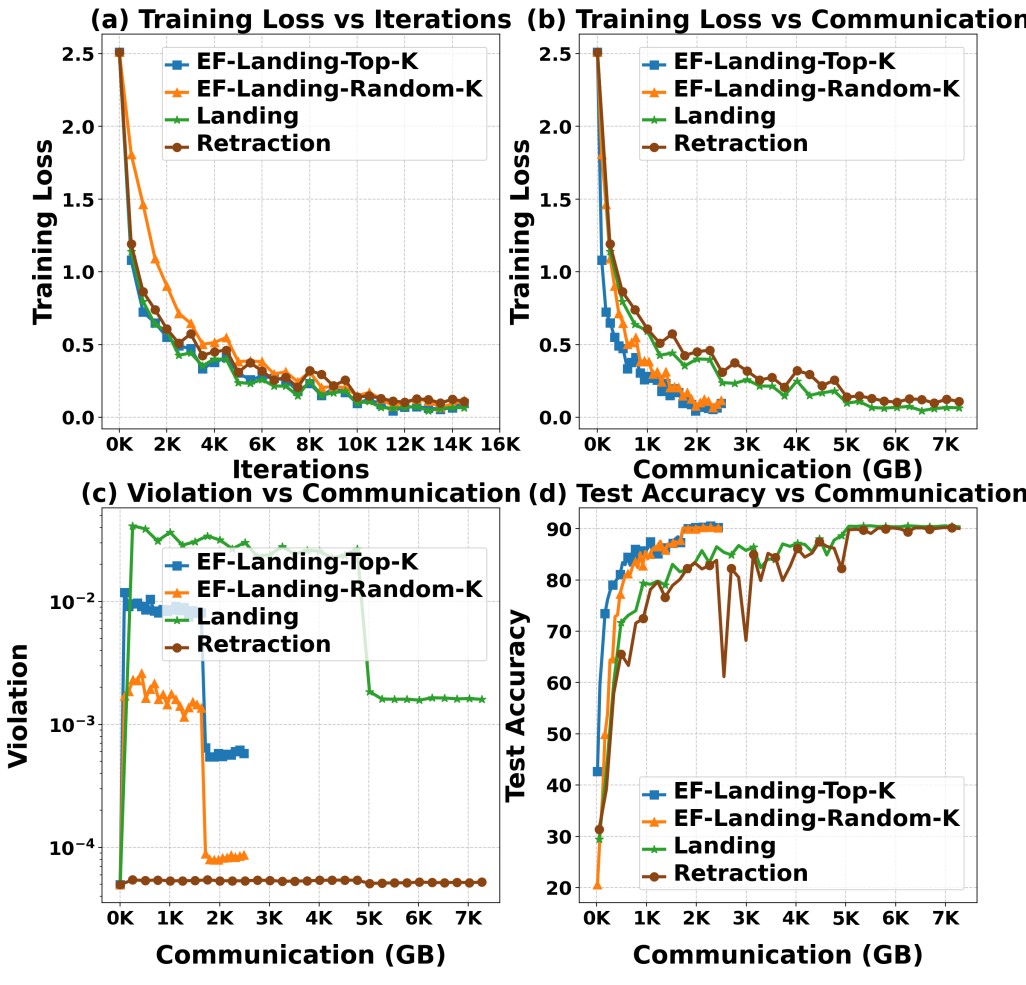

Figure 9: Performance of EF-Landing and other algorithms on deep learning with VGG16 on CIFAR-10.

As shown in Figures 3 and 9, in terms of satisfying the constraint conditions, the EF-Landing algorithm not only reduces $\mathcal{N}(X)$ with less communication overhead compared to the Landing algorithm, but also satisfies the constraints more accurately. Furthermore, in terms of computational cost, the EF-Landing algorithm offers a clear advantage over the QR retraction method, being free of QR decomposition calculations.

| Method/Hyperparameter | $\gamma$ | $\eta$ | Clipping | $\lambda$ | compress rate |
|---|---|---|---|---|---|
| EF-Landing-Top-$K$ | 0.1 | 0.5 | $10^8$ | 1.0 | 0.2 |
| EF-Landing-Random-$K$ | 0.1 | 0.5 | $10^8$ | 1.0 | 0.2 |
| Landing | 0.1 | – | – | 1.0 | – |
| Retraction | 0.1 | – | – | 1.0 | – |

Table 8: Hyperparameters: ResNet-18 on CIFAR-10

## D.3. Comparison with the Penalty method

This section mainly compares the performance of the EF-Landing algorithm and the Penalty method.

### D.3.1. PCA PROBLEM

The data generation process for the Online PCA problem follows the description provided in Appendix D.1.1. The step size $\gamma$ is selected via grid search over the set $\{0.001, 0.01, 0.1, 1.0\}$. As shown in Figures 10 to 13, compared with the EF-Landing-based algorithm, the Penalty method is highly sensitive to the choice of the penalty parameter $\lambda$. When $\lambda$ is set too small, the orthogonality constraint cannot be effectively enforced. On the other hand, choosing a larger $\lambda$ may introduce numerical instability, necessitating a smaller step size and consequently slowing down convergence. In contrast, the EF-Landing-based algorithm can satisfy the orthogonality constraint with a relatively small $\lambda$, leading to faster convergence, and consistently achieves a smaller Riemannian gradient norm.

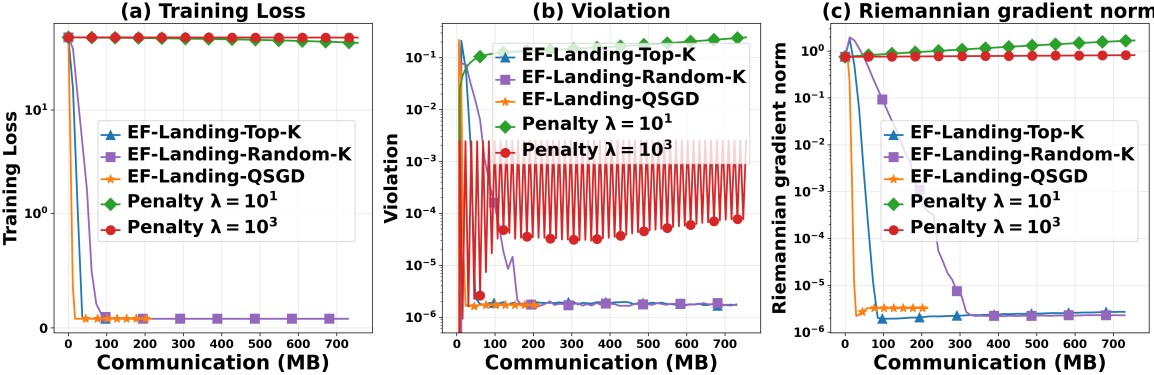

Figure 10: Comparison with Penalty Method: $p = 100$

| Method/Hyperparameter | $\gamma$ | Clipping | $\lambda$ | compress rate /s |
|---|---|---|---|---|
| EF-Landing-Top-$K$ | 1.0 | $10^8$ | 0.5 | 0.1 |
| EF-Landing-Random-$K$ | 1.0 | $10^8$ | 0.5 | 0.1 |
| Penalty method | 0.01 | – | 10.0 | – |
| Penalty method | 0.001 | – | 1000.0 | – |

Table 9: Hyperparameters: Comparison with Penalty Method $p = 100$

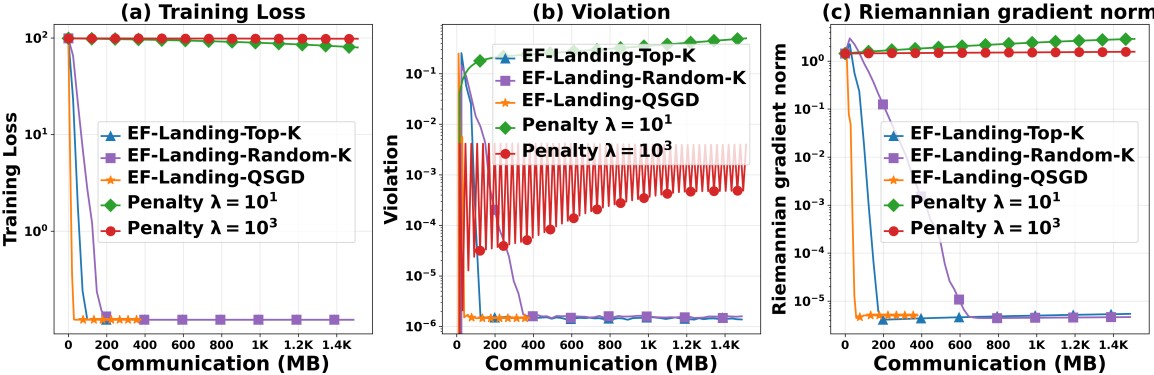

Figure 11: Comparison with Penalty Method: $p = 200$

| Method/Hyperparameter | $\gamma$ | Clipping | $\lambda$ | compress rate /s |
|---|---|---|---|---|
| EF-Landing-Top-$K$ | 1.0 | $10^8$ | 0.5 | 0.1 |
| EF-Landing-Random-$K$ | 1.0 | $10^8$ | 0.5 | 0.1 |
| Penalty method | 0.01 | – | 10.0 | – |
| Penalty method | 0.001 | – | 1000.0 | – |

Table 10: Hyperparameters: Comparison with Penalty Method $p = 200$

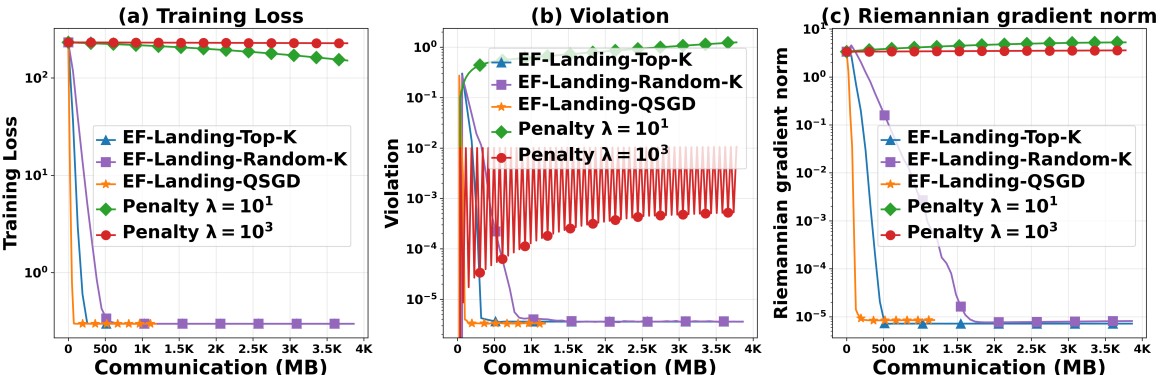

Figure 12: Comparison with Penalty Method: $p = 500$

| Method/Hyperparameter | $\gamma$ | Clipping | $\lambda$ | compress rate /s |
|---|---|---|---|---|
| EF-Landing-Top-$K$ | 1.0 | $10^8$ | 0.5 | 0.1 |
| EF-Landing-Random-$K$ | 1.0 | $10^8$ | 0.5 | 0.1 |
| Penalty method | 0.01 | – | 10.0 | – |
| Penalty method | 0.001 | – | 1000.0 | – |

Table 11: Hyperparameters: Comparison with Penalty Method $p = 500$

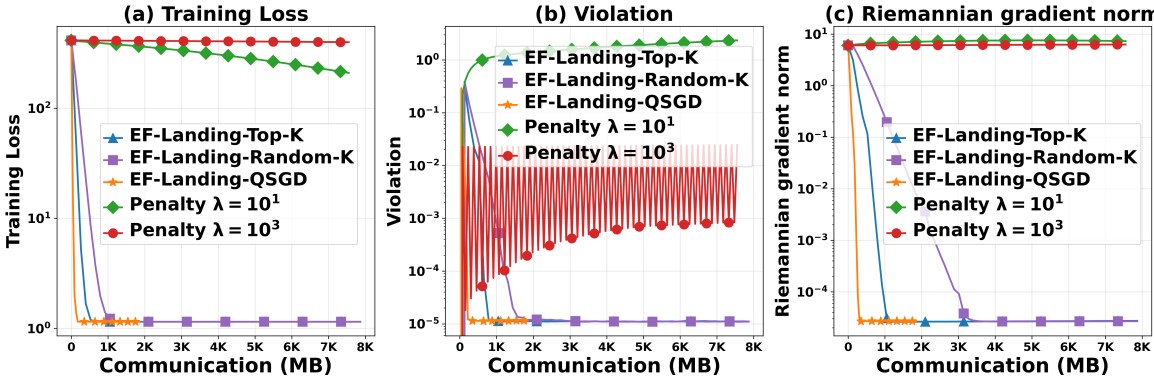

Figure 13: Comparison with Penalty Method: $p = 1000$

| Method/Hyperparameter | $\gamma$ | Clipping | $\lambda$ | compress rate /s |
|---|---|---|---|---|
| EF-Landing-Top-$K$ | 1.0 | $10^8$ | 0.5 | 0.1 |
| EF-Landing-Random-$K$ | 1.0 | $10^8$ | 0.5 | 0.1 |
| Penalty method | 0.01 | – | 10.0 | – |
| Penalty method | 0.001 | – | 1000.0 | – |

Table 12: Hyperparameters: Comparison with Penalty Method $p = 1000$

### D.3.2. NEURAL NETWORK

We compare the Penalty method and the EF-Landing-based algorithm on the CIFAR-10 dataset using VGG16 and ResNet-18 neural network architectures. A grid search was conducted over $\gamma \in \{0.001, 0.01, 0.1, 1.0\}$ and $\eta \in \{0.1, 0.5, 0.7\}$ to determine the optimal hyperparameters. Each algorithm was trained for 150 epochs, and after the 100th epoch, $\gamma$ was reduced to $\frac{1}{10}$ of its original value. A batch size of 128 was used throughout the training process.

From the experimental results (Figures 14,15), we observe that the EF-Landing-based algorithm achieves higher accuracy while significantly reducing communication costs, and it satisfies the orthogonality constraint more precisely. In contrast, the Penalty method performs poorly. When using a relatively small penalty coefficient, such as $\lambda = 10$, it can achieve relatively high accuracy, but fails to satisfy the constraint as precisely as the EF-Landing-based algorithm. On the other hand, increasing the penalty to $\lambda = 1000$ improves constraint satisfaction but leads to a significant drop in accuracy.

| Method/Hyperparameter | $\gamma$ | $\eta$ | Clipping | $\lambda$ | compress rate |
|---|---|---|---|---|---|
| EF-Landing-Top-$K$ | 0.1 | 0.5 | $10^8$ | 1.0 | 0.2 |
| EF-Landing-Random-$K$ | 0.1 | 0.1 | $10^8$ | 1.0 | 0.2 |
| Penalty method | 0.01 | – | – | 10.0 | – |
| Penalty method | 0.001 | – | – | 1000.0 | – |

Table 13: Hyperparameters: EF-Landing Algorithm and Penalty Method: VGG16 on CIFAR-10

| Method/Hyperparameter | $\gamma$ | $\eta$ | Clipping | $\lambda$ | compress rate |
|---|---|---|---|---|---|
| EF-Landing-Top-$K$ | 0.1 | 0.5 | $10^8$ | 1.0 | 0.2 |
| EF-Landing-Random-$K$ | 0.1 | 0.5 | $10^8$ | 1.0 | 0.2 |
| Penalty method | 0.01 | – | – | 10.0 | – |
| Penalty method | 0.001 | – | – | 1000.0 | – |

Table 14: Hyperparameters: EF-Landing Algorithm and Penalty Method: ResNet-18 on CIFAR-10

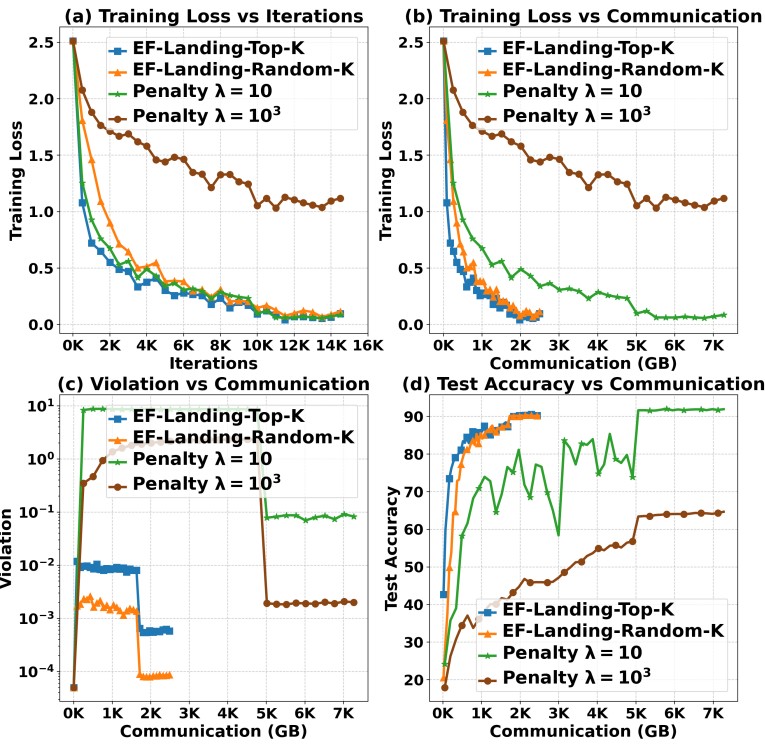

Figure 14: Comparison of EF-Landing Algorithm and Penalty Method: VGG16 on CIFAR-10

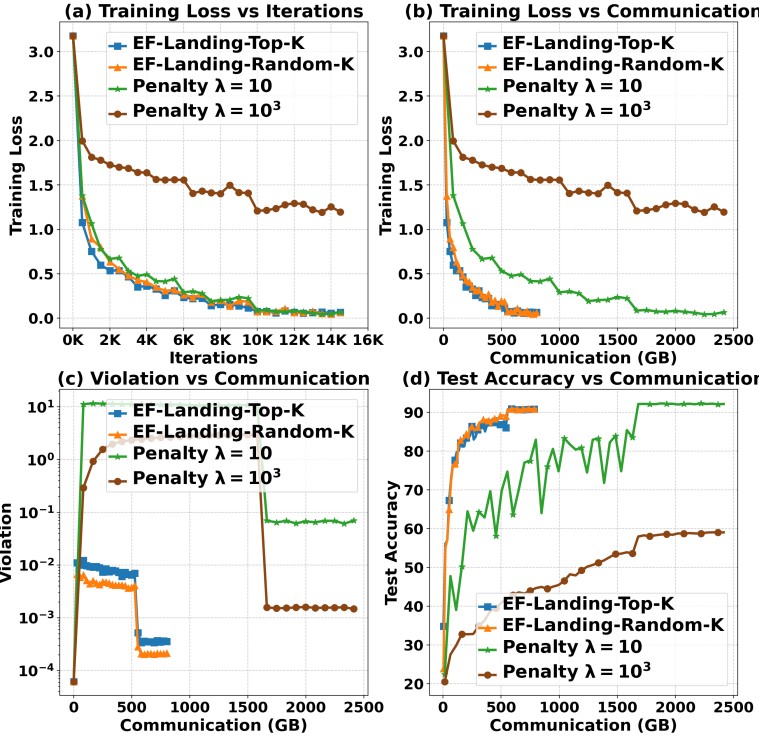

Figure 15: Comparison of EF-Landing Algorithm and Penalty Method: ResNet-18 on CIFAR-10

### D.4. Comparison with Decentralized Setting

We performed online PCA experiments using synthetic datasets, each comprising 1000 data points per node with $n = 100$ and $p = 5$, following a data generation process which is detailed in Appendix D.1.1.

For the Decentralized distributed experiment, we chose the DRSGD and DRGTA algorithms from (Chen et al., 2021) as baselines. In the experiment, we set the number of nodes to 20, the communication rounds $t$ to 1, and selected a ring topology with the Metropolis constant matrix associated with the graph. Under a ring topology communication structure, each node interacts solely with its two immediate neighbors. When the number of nodes is set to 20, this configuration is approximately equivalent to the centralized setting with a compression rate of 0.1. For DRGTA, we set $\hat{\beta}$ to 0.05, while for the DRSGD algorithm, we set $\hat{\beta}$ to 0.01.

For the EF-Landing algorithm, we set $\lambda = 1.0$, the learning rate $\gamma = 0.2$, and the compress rate to 0.1. It is worth noting that the metric used to evaluate the satisfaction of the manifold constraint follows the approach proposed in this paper, which is based on the canonical correlations (Golub & Zha, 1995) between $X_k$ and $X_*$.

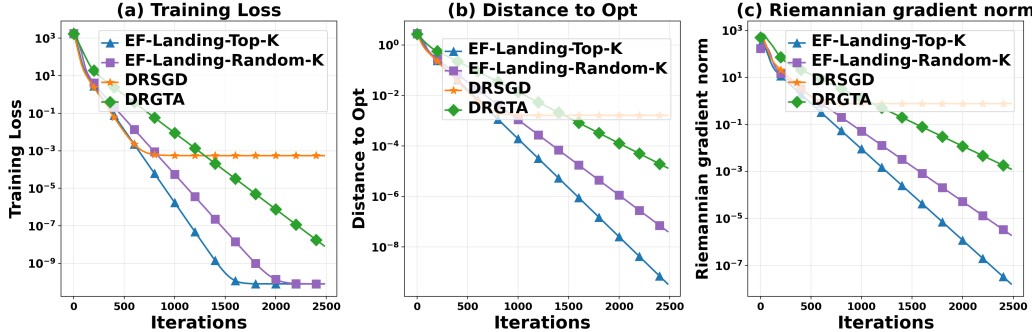

Figure 16: Comparison of EF-Landing Algorithm and Decentralized Algorithm On Synthetic Dataset 1

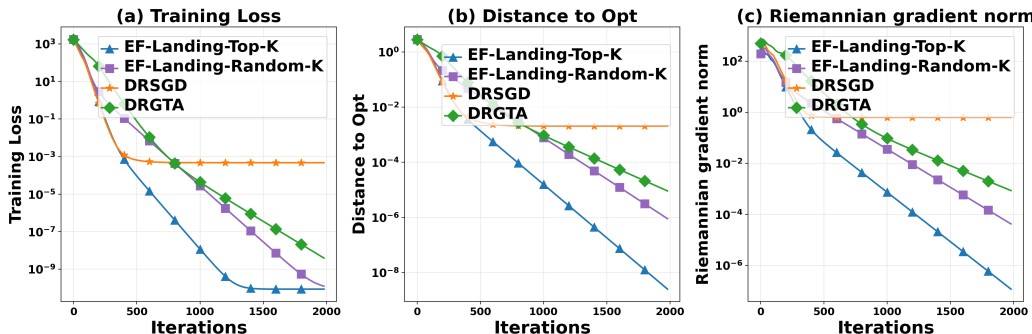

Figure 17: Comparison of EF-Landing Algorithm and Decentralized Algorithm On Synthetic Dataset 2

Figures 16 and 17 demonstrate that the EF-Landing-based algorithm achieves faster convergence compared to DRGTA and DRSGD, while also exhibiting superior constraint satisfaction performance.

### D.5. Comparison with Centralized Multi-Local-Update Setting

Using the algorithm proposed in (Zhang et al., 2024) as a benchmark, we conducted experiments on the Online PCA problem using synthetic datasets. The number of nodes was set to 4, with a total sample size of $N = 800$, $n = 20$ or $n = 200$, and $p = 5$.

For Algorithm 1 proposed in (Zhang et al., 2024), we set the parameters as $\tau = 5, 10$, $\eta = 0.001$, and $\eta_g = 1.0$, where $\tau$ represents the number of local update rounds, $\eta$ denotes the learning rate for local updates, and $\eta_g$ represents the learning rate at the center.

For the EF-Landing algorithm, we set $\lambda = 1.0$, the learning rate $\gamma = 0.2$, and the compress rate to 0.1.

Figures 18 and 19 demonstrate that the EF-Landing-based algorithm exhibits superior performance in terms of constraint violation and achieves comparable convergence speed to the benchmark.

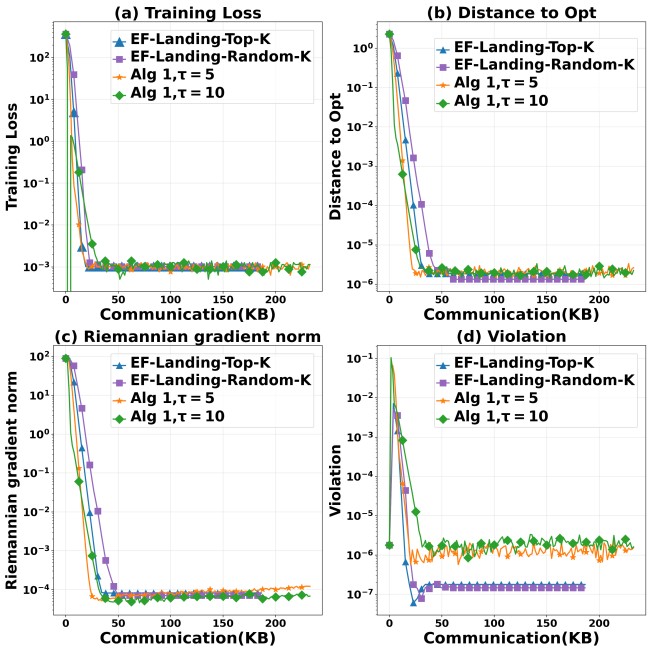

Figure 18: Comparison of EF-Landing Algorithm and Algorithm 1 in [Zhang et al. 2024], synthetic dataset $n = 20$

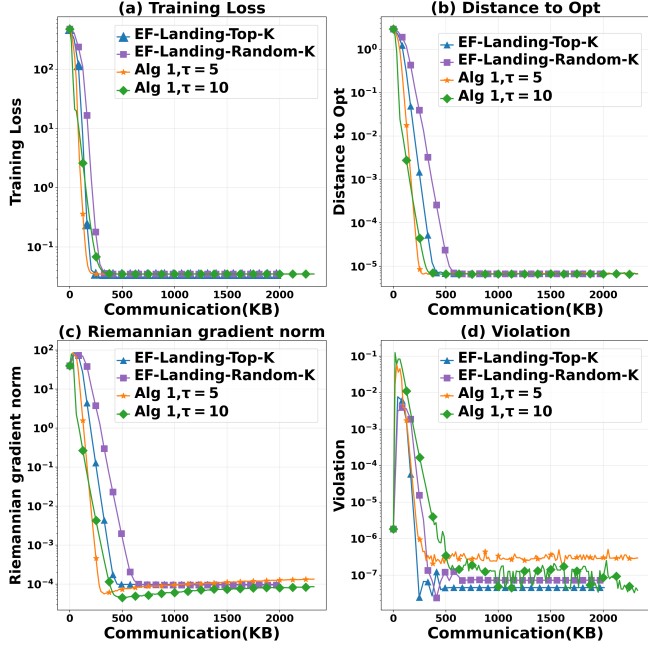

Figure 19: Comparison of EF-Landing Algorithm and Algorithm 1 in [Zhang et al. 2024], synthetic dataset $n = 200$

