# OpenReview forum: "Distributed Retraction-Free and Communication-Efficient Optimization on the Stiefel Manifold"
_ICML.cc/2025/Conference — ICML 2025 poster_

### Official Review · Reviewer_ytVq · 2025-03-10

**Overall Recommendation:** 4

**Summary:**

The paper introduces EF-Landing, a novel distributed optimization algorithm for stochastic optimization on the Stiefel manifold. EF-Landing is retraction-free and communication-efficient, incorporating gradient compression and error feedback mechanisms. The authors establish sharp convergence guarantees and demonstrate that EF-Landing achieves the same asymptotic linear speedup as existing methods without communication compression. The paper also generalizes EF-Landing to block-wise Stiefel manifolds, enhancing its applicability to structured constraints. Extensive numerical experiments validate the theoretical findings.

**Claims And Evidence:**

The main claims of the paper are:
1.	EF-Landing is the first retraction-free and communication-efficient algorithm for distributed stochastic optimization on the Stiefel manifold.
2.	The algorithm ensures convergence while significantly reducing communication overhead.
3.	The convergence rate of EF-Landing matches that of existing methods without communication compression.
4.	The method generalizes to block-wise Stiefel manifolds, extending its practical applicability.
The claims are well-supported by theoretical proofs and numerical experiments. The convergence guarantees are rigorously derived, and extensive experiments confirm that EF-Landing performs comparably to existing methods while reducing communication costs. The inclusion of error feedback ensures stability and accuracy despite compression.

**Essential References Not Discussed:**

N/A

**Experimental Designs Or Analyses:**

The experimental design is sound. The authors provide experiments on two groups of problems: the distributed online PCA for deterministic scenario and deep learning using VGG16 neural network architecture with orthogonal constraints applied to the convolutional layers for stochastic scenario, comparing EF-Landing with other algorithms for optimization on Stiefel manifold, including vanilla Landing, QR retraction. The experiments are well-structured, demonstrating the effectiveness of EF-Landing in terms of both convergence rate and communication efficiency.

**Methods And Evaluation Criteria:**

The proposed methods and evaluation criteria are appropriate for the problem. The authors use standard theoretical tools from optimization on manifolds to establish the convergence guarantees and rates for EF-Landing. The experimental evaluation includes benchmark problems such as distributed online PCA and deep learning tasks. The comparison with existing methods, including QR retraction and vanilla Landing methods, provides a solid basis for evaluating EF-Landing's effectiveness.

**Other Comments Or Suggestions:**

•	Expanding experiments to additional machine learning tasks could further strengthen the empirical validation.

**Other Strengths And Weaknesses:**

Strengths:
•	The paper provides sharp theoretical convergence guarantees for EF-Landing.
•	The convergence rate of EF-Landing matches that of existing methods without communication compression.
•	The method generalizes to block-wise Stiefel manifolds, extending its practical applicability.

Weaknesses:
•	Experimental evaluation could be expanded to include more diverse datasets and application scenarios.

**Questions For Authors:**

N/A

**Relation To Broader Scientific Literature:**

The paper is well-situated within the broader literature on optimization on manifolds, distributed optimization, and communication-efficient training. It builds on prior work in these areas and makes a novel contribution by integrating error feedback with retraction-free optimization. The references to previous work, such as the Landing method and communication-efficient distributed optimization techniques, are appropriate.

**Theoretical Claims:**

The theoretical claims of convergence analysis are well-supported by rigorous proofs. The convergence analysis begins by presenting the necessary conditions, then proceeds to establish the convergence guarantees and rates for EF-Landing. The authors present the standard assumptions for optimization on the Stiefel manifold, supporting lemmas and main convergence theorem. Besides, the authors establish convergence rate in deterministic scenario and stochastic scenario.

---

> ### Author Rebuttal · Authors · 2025-04-01
>
> We sincerely thank the reviewer for their insightful feedback and constructive comments. Below, we reiterate our novel contributions which the reviewer had mentioned, provide other theoretical innovations for reference, and show the plan of expanding follow-up experiments.
>
> **1. Reiteration of Contributions.**
>
> - Sharp convergence analysis: Instead of analyzing convergence results case-by-case for different settings, we provided a general convergence result. It can reduce to various specific situations (deterministic or stochastic scenarios, with or without compression, with or without momentum) by choosing different auxiliary constants. When deterministic setting without compression or momentum is chosen, our result exactly corresponds to the result in [Ablin et al. 2024].
>
> - Block-wise generalization: Prior work for deep learning assumes fully vectorizable variables, failing to address block-wise constraints (e.g., orthogonality in neural network layers). Our work, however, directly tackles block-wise structures (Section 6), proving convergence under a unified step size (Theorem K.1). To our knowledge, this is the first analysis for block-wise orthogonal optimization, addressing practical deep learning architectures.
>
> - Extensive experiments: Prior work concerned with optimization on Stiefel manifolds mainly focuses on traditional problems like PCA. We reviewed the more general application of orthogonal constraints to deep learning, and conducted extensive experiments on diverse deep learning tasks. Brilliant results demonstrated the applicability and efficiency of our algorithm.
>
>
> **2. Other Theoretical Innovations.**
>
> Apart from the insights the reviewer had mentioned, we also summarize more of our technical contributions below. We hope these align with the reviewer’s expectation.
>
> **(a) Reduced Assumptions.**
> - Prior analyses of the Landing method [Ablin et al. 2024] require restrictive assumptions: bounded local Riemannian gradients, explicit bounds on intermediate symmetric matrices, and unbiasedness conditions for specific Riemannian gradients.
> - **Our Work:** We eliminate these constraints, relying only on standard smoothness and mild gradient bounds (Assumptions 3.1–3.2). This broader generality enables more applications.
>
> **(b) Perturbation-Tolerant Analysis.**
> - Existing Landing convergence guarantees fail under gradient perturbations (e.g., compressed gradients). While trivial in Euclidean settings, perturbations on the Stiefel manifold introduce non-negligible geometric distortion.
> - **Our Work:** We rebuild the analysis from first principles, introducing a **perturbation-compatible merit function** (Lemma 5.6) that rigorously accounts for compression errors. This is the first provably robust Landing variant for compressed gradients (Theorem 5.7).
>
> **(c) Gradient Clipping for Safety Guarantees.**
> - Compressed gradients only remain within the safety region in expectation, risking constraint violations.
> - **Our Work:** A novel clipping strategy enforces deterministic safety without introducing bias.
>
> **3. Additional experiments.**
>
> All newly added experiments can be found in the **Additional Result Sheet (ARS)** https://anonymous.4open.science/r/EF-Landing-B6E4. Figures 5 & 6 add the comparison of EF-Landing Algorithm and Penalty Method using VGG16 and ResNet18 on CIFAR-10, which further shows the efficiency of our algorithm. Moreover, we are expected to conduct more experiments, where more diverse datasets, more application scenarios, comparisons among more relevant methods are all taken into consideration.
>
> We again appreciate the reviewer for the valuable feedback and for recognizing the contributions of our work.

---

### Official Review · Reviewer_kbBP · 2025-03-14

**Overall Recommendation:** 3

**Summary:**

This paper introduces EF-Landing, a retraction-free and communication-efficient algorithm for distributed stochastic optimization on the Stiefel manifold.

**Claims And Evidence:**

The paper's main claims regarding retraction-free optimization, communication efficiency, and error feedback improving convergence are generally supported by theoretical analysis and empirical results.

**Essential References Not Discussed:**

N/A

**Experimental Designs Or Analyses:**

Within the current distributed setting, it would be beneficial to include a comparison with multiple local updates.

Consider evaluating against methods that incorporate local updates, such as those in Zhang et al. (2024), NeurIPS, which study nonconvex federated learning on compact smooth submanifolds with heterogeneous data.

Zhang, J., Hu, J., So, A.M.C. and Johansson, M., 2024. Nonconvex federated learning on compact smooth submanifolds with heterogeneous data. Advances in Neural Information Processing Systems, 37, pp.109817-109844.

**Methods And Evaluation Criteria:**

The proposed method is designed for distributed systems and reduces communication via compression. However, it relies on centralized coordination, requiring communication between worker nodes and the master at every iteration, which may limit communication efficiency. A more decentralized approach or a method allowing multiple local iterations before synchronization could further improve efficiency.

**Other Comments Or Suggestions:**

**Equation (2):** Please validate the expression of the Riemannian gradient $\textrm{grad} f(X)$ by explicitly specifying the associated Riemannian metric used in the derivation.

**Algorithm 1:** Is the gradient clipping with constant $L'$ necessary? If so, how should $L'$ be estimated in practice? Please clarify its role in ensuring convergence.

**Other Strengths And Weaknesses:**

A key weakness is that the main theorems and lemmas largely follow existing results without significant technical difficulty, limiting the novelty of the theoretical contributions. From an algorithmic perspective, the current distributed setting is somewhat outdated, as modern approaches often consider decentralized communication or centralized frameworks with multiple local updates before aggregation.

**Questions For Authors:**

N/A

**Relation To Broader Scientific Literature:**

The paper's main contribution lies in merging communication compression with a retraction-free optimization method. The theoretical analysis primarily follows from error feedback techniques to control errors in the Euclidean gradient, which are then incorporated into existing retraction-free analyses. As a result, the work mainly applies existing technical tools rather than introducing new theoretical insights. Additionally, the considered distributed setting is relatively simple, and extending the approach to more complex decentralized or federated settings would enhance its impact.

**Theoretical Claims:**

The proofs seem fine.

---

> ### Author Rebuttal · Authors · 2025-04-01
>
> We sincerely thank the reviewer for their insightful feedback and constructive comments. Below, we address each point in detail. All newly added experiments can be found in the **Additional Result Sheet (ARS)** https://anonymous.4open.science/r/EF-Landing-B6E4
>
> **1. Distributed learning**
>
> Decentralized Learning (DL), Federated Learning (FL), and Compressed Learning (CL) represent three **orthogonal** research directions in communication-efficient distributed learning. Specifically, DL focuses on *which neighbors to communicate with* (topology design), FL addresses *when to communicate* (synchronization frequency), and CL determines *what to communicate* (data/gradient compression).
>
> Given that these approaches optimize along fundamentally different axes, directly comparing their efficiency or asserting the superiority of one over another might not be appropriate. Each branch presents unique theoretical and practical challenges. Thus, rather than being considered "outdated," each methodology remains relevant depending on the specific problem constraints and system requirements.
>
> **2. Comparison with DL**
>
> While it is challenging to determine whether CL or DL is superior in general scenarios, we conduct experiments to compare them in specific settings, where we choose proper topology for DL and compression rate for CL to make the communication quantity per iteration equally matched. Figures 7 & 8 in ARS demonstrate that EF-Landing achieves slightly higher communication efficiency than decentralized manifold methods DRSGD and DRGTA [Chen et al. 2021].
>
> **3. Comparison with FL**
>
> We appreciate the reviewer for bringing this valuable reference to our attention. We will include it in the related work. Below, we provide a comparison between the two approaches:
>
> - Communication paradigm: Zhang et al. reduce communication overhead through multiple local steps, whereas EF-Landing achieves this via gradient compression. Which approach is more efficient depends on the specific application.
> - Computational overhead: Zhang et al. rely on manifold projection, which can be computationally expensive, while EF-Landing employs retraction-free methods that involve only matrix products.
> - Addressing data heterogeneity: Zhang et al. introduce a correction step to mitigate data heterogeneity, whereas EF-Landing leverages error feedback for correction.
> - Convergence: Zhang et al. do not converge exactly to the stationary solution but only to a neighborhood around it, whereas EF-Landing achieves exact convergence.
>
> Furthermore, we perform additional experiments to compare EF-Landing with Zhang et al. Figures 9 & 10 in ARS show that their performances are roughly equally matched in terms of communication quantities.
>
> **4. Theoretical innovations.**
>
> We summarize our technical contributions below.
>
> **(a) Reduced Assumptions**
> - Prior analyses of the Landing method [Ablin et al. 2024] require restrictive assumptions: bounded local Riemannian gradients, bound on an intermediate symmetric matrix, and unbiasedness conditions for specific Riemannian gradients.
> - **Our Work:** We eliminate these constraints, relying only on standard smoothness and mild gradient bounds (Assumptions 3.1–3.2). This broader generality enables more applications.
>
> **(b) Perturbation-Tolerant Analysis**
> - Existing Landing convergence guarantees fail under gradient perturbations (e.g., compressed gradients). While trivial in Euclidean settings, perturbations on the Stiefel manifold introduce non-negligible geometric distortion.
> - **Our Work:** We rebuild the analysis from first principles, introducing a **perturbation-compatible merit function** (Lemma 5.6) that rigorously accounts for compression errors. This is the first provably robust Landing variant for compressed gradients (Theorem 5.7).
>
> **(c) Block-Wise Orthogonal Constraints**
> - Prior work assumes fully vectorizable variables, failing to address block-wise constraints (e.g., orthogonality in NN layers).
> - **Our Work:** We directly tackle block-wise structures (Section 6), proving convergence under a **unified step size** (Theorem K.1). To our knowledge, this is the first analysis for block-wise orthogonal optimization, addressing practical deep learning architectures.
>
> **(d) Gradient Clipping for Safety Guarantees**
> - Compressed gradients only remain within the safety region in expectation, risking constraint violations.
> - **Our Work:** A novel clipping strategy enforces deterministic safety without introducing bias.
>
> **5. Other Comments.**
> - We use the canonical metric of the Stiefel manifold.
> - The necessity of gradient clipping stems from the fact that contractive compressors only guarantee contraction in expectation. There is a small risk of gradient explosion after compression. To ensure the iteration remains within a safe region, we assume bounded gradients (It is fine to overestimate it), with which clipping ensures safety without introducing additional noise.

---

> > ### Comment · Reviewer_kbBP · 2025-04-02
> >
> > I appreciate the reviewer’s thoughtful responses and the addition of numerical tests, which address my concerns. I am happy to raise my rating.

---

> > > ### Author Response · Authors · 2025-04-05
> > >
> > > We sincerely thank the reviewer for their thoughtful engagement and for considering our clarifications and insights. We greatly appreciate their willingness to update their review based on our response and additional experiments. Their constructive feedback and supportive remarks are very encouraging and helpful in strengthening our work.

---

### Official Review · Reviewer_b4uN · 2025-03-25

**Overall Recommendation:** 3

**Summary:**

Paper provides an error feedback based algorithm to solved distributed optimization problem on Stiefel manifold (set of orthonormal matrices). This algorithm generalizes recently proposed retraction-free Landing method to a low-comm complexity distributed algorithm. Authors also provide theoretical convergence results and empirical evidence to support their claims. Authors also extend their work to problems with block-level orthonormality property.

**Claims And Evidence:**

Most of theoretical claims seem correct. See below for concerns about motivation and experiments.

**Essential References Not Discussed:**

Seem alright. Not familiar with literature enough.

**Experimental Designs Or Analyses:**

I don’t fully grasp why penalty method fails here.

1. I have concerns of why more details about the penalty method is not provided. Both EF-Landing and penalty method (of course with appropriate compression and error feedback) have the a regularization weight \lambda which needs to be tuned.
2. I would also like to penalty method for the NN result plots. Not sure why they are omitted.
3. Why is penalty method objective being omitted? Even if the values are in different range it is useful to provide it (at least in appendix if space is limited)

**Methods And Evaluation Criteria:**

1. How is number of violations defined? This is crucial to understand the claims.
2. How are the hyper parameters for each of the methods tuned? No details are provided to improve confidence in results.
3. No comparison empirical comparison against these baselines [Chen et al., 2021], [Wang et al., 2022], [Qu et al., 2024], [Zhao et al., 2025]

**Other Comments Or Suggestions:**

Minor:
1. Assumption 3.2: better to use a different variable than X to avoid confusion with variable of (1) and argument of compressor.

**Other Strengths And Weaknesses:**

I have non-trivial concerns about significance and novelty of the work. Most of techniques used are known apriori and this works seems straight forward combinations of them.

1. It is not super clear how landing improves over standard penalty/regularization based methods. This wasn’t discussed in details, and its convergence rate was not compared against. As far as I understand there always exists some value of regularization (penalty) weight that lead to similar approximate constraint satisfaction. Also see other sections concerns about the empirical evidence of same.

2. Another concern is that the failure of compression without error feedback is not special to Stiefel manifold as implied by the authors. Same argument for example work even convex constraints (disc instead of circle) [Richtarik et al., 2021]. It even is known for unconstrained problems [Seide et al., 2014]. So it is slightly misleading for authors to omit these prior knowledge.

3. While it is insightful that compressing the descent direction directly could make the two sub fields non-orthogonal, it is also natural to compute gradient of the penalty term at the centralized server as it is not client-data dependent. This is a standard known practice for regularizer in decentralized optimization. Authors omit this in their discussion of their choice.

Due above concerns and lack of elaboration by authors, I don’t understand what is the novelty of the work.

**Questions For Authors:**

Please see above. Summarizing:
1. Advantage over penalty method both theoretically and empirically
2. Novelty of the technical ideas and significance of work

**Relation To Broader Scientific Literature:**

See other strengths and weaknesses

**Theoretical Claims:**

Proofs seem correct. Most of the techniques and ideas used in the paper are already known and hence straight-forward.

---

> ### Author Rebuttal · Authors · 2025-04-01
>
> We sincerely thank the reviewer for their insightful feedback and constructive comments. Below, we address each point in detail. All newly added experiments can be found in the **Additional Result Sheet (ARS)** https://anonymous.4open.science/r/EF-Landing-B6E4
>
> **1. Landing v.s. Penalty**
>
> The Landing method can be roughly analogous to the Augmented Lagrangian method, while the Penalty method corresponds to traditional penalty approaches in constrained optimization. Below, we provide a detailed comparison.
>
> **(a). Traditional Penalty Methods: Fundamental Trade-offs**
>    - **Small $\lambda$ Regime**:
>      - Pros: Preserves fast convergence toward optimality.
>      - Cons: Fails to enforce constraint feasibility.
>    - **Large $\lambda$ Regime**:
>      - Pros: Easy to ensure strict feasibility.
>      - Cons: Induces ill-conditioning problems and slows convergence.
>    - **Practical Implementation**:
>      - Delicate schedule-based tuning (e.g., slowly increasing $\lambda$) is necessary to balance these competing goals. Even then, achieving both fast feasibility and optimality is non-trivial and problem-dependent in practical experiments.
>
> **(b) Landing Method: Decoupling the Trade-off**
>
> - **Key Advantage**: Employs a *constant, moderate $\lambda$* (typically $\lambda = 1$) to **simultaneously** ensure both optimality and feasibility.
> - **Benefits**:
>   - Avoids the ill-conditioning issues associated with large $\lambda$ in penalty methods.
>   - Eliminates the need for problem-specific $\lambda$ selection and extensive parameter tuning.
>
> While both Landing and Penalty methods achieve the same asymptotic convergence rate, Landing demonstrates superior experimental performance due to the above benefits. Our new experiments in Figures 1 - 4 in ARS validate this conclusion.
>
> **2. Error Feedback (EF)**
>
> - **Review of Prior Work.** Previous studies [Richtárik et al., 2021; Seide et al., 2014] have shown that vanilla gradient compression using contractive compressors fails to converge in distributed learning when **data heterogeneity** is present. In such cases, EF is necessary to address this issue and ensure convergence.
>
> - **Unique Challenges in Manifold Optimization.** Our findings reveal—for the first time—that even in a deterministic single-node setting (where no data heterogeneity exists and vanilla gradient compression is expected to converge)—vanilla gradient compression unexpectedly fails when a Stiefel manifold constraint is imposed. This phenomenon highlights a fundamental distinction in manifold-constrained optimization: unlike unconstrained settings where single-node compression succeeds without error feedback, the Stiefel manifold’s geometry introduces additional barrier to gradient compression.
>
> While we carefully reviewed [Richtárik et al., 2021], we were unable to locate a discussion of convex constraint sets (e.g., discs). We would greatly appreciate it if the reviewer could direct us to the specific page or section on this point. We can expect that problems with disc constraint may have similar trouble like ours, but this trouble is more crucial for our settings because Stiefel manifold has no unconstrained interior points.
>
>
> **3. Compute Penalty Gradient at Server**
>
> We thank the reviewer for the valuable insights drawn from decentralized optimization. We agree that computing client-independent terms, such as the penalty term, at the centralized server can also provide motivation for certain aspects of our algorithmic design, and we will include a detailed discussion of this in the revision. However, the central aspect of our analysis is that the orthogonality property, established in Proposition 4.1, is crucial for ensuring convergence guarantees. While the algorithm can indeed be motivated from different perspectives, the convergence guarantees cannot be established without the above orthogonality property.
>
> **4. Theoretical Novelties**
>
> Apart from the above clarifications, we summarize other theoretical novelties in our response 4 to Reviewer kbBP.
>
> **5. Other Comments**
>
> - More experiments. Figures 7 & 8 in ARS show comparison with decentralized manifold methods DRSGD and DRGTA. Detailed analysis can be found in our response 1 & 2 to Reviewer kbBP. Figures 1 - 6 in ARS show comparison with penalty method for PCA and NN tasks. It is observed that penalty method fails to reach optimality and feasibility when using a fixed penalty parameter, and it also performs not so good in NN tasks due to the above trade-offs.
> - Violation. The penalty term $\frac14\\|X^\top X − I_p\\|^2$ implies the magnitude of the violation.
> - Hyperparameters. All penalty parameters for Landing are set to 1, and the momentum for the NN experiments is set to 0.9. Step sizes are selected on a case-by-case basis through grid search.
> - We used the notation $X$ to define compressor considering that decision variable and its gradient are of the same shape. But the reviewer’s suggestion is helpful and we will revise it.

---

> > ### Comment · Reviewer_b4uN · 2025-04-09
> >
> > I thank the authors for their response to my comments and the new experimental results. These, have partially addressed some of my concerns like the advantage of landing method, and comparison to more baselines. I am also glad to hear that the authors plan to add more experimental results. I agree with this sentiment and further believe that this paper requires a considerable revision to incorporate all these arguments in detail including discussion of technical novelty, landing method vis-a-vis penalty method, new experiments, and its relation to prior work. So I will change my recommendation from weak reject to weak accept.
> >
> > I apologize for the confusion regarding referencing [Richtarik et al., 2021]. I was pointing out Sec 2.2 of the paper which lists known compression failure results and provides a summary. For example, [Karimireddy et al., 2019] provides counter examples where compressed SGD fails even when there is no “data heterogeneity.” Authors seem to have missed this paper in their draft. I am glad that the authors see that replacing Stiefel manifold with a disc constraint will also lead to similar failure of compressed projected SGD. This proves that general constrained optimization fail under compression without error feedback and it is not just a special consequence of optimizing on Stiefel manifold. To be clear, I don’t have any follow up ask here.
> >
> > [Karimireddy et al., 2019] Sai Praneeth Karimireddy, Quentin Rebjock, Sebastian Stich, and Martin Jaggi. Error feedback fixes SignSGD and other gradient compression schemes. In 36th International Conference on Machine Learning (ICML)

---

> > > ### Author Response · Authors · 2025-04-09
> > >
> > > We sincerely thank the reviewer for their kindness to review our response and new experiments. We also greatly appreciate their thoughtful and constructive suggestions.
> > >
> > > Our paper will be properly revised based on the reviewer’s comments:
> > > - The section of contributions will be reorganized to better highlight our theoretical novelty.
> > > - Additional discussion of Landing compared with other methods (such as penalty methods) will be included to emphasize Landing’s advantages.
> > > - Results of further experiments, including comparisons with baseline methods in decentralized settings, will be added to illustrate the competitiveness of our approach.
> > > - More relevant prior work, such as [Karimireddy et al., 2019] will be cited to clarify the necessity of error feedback from a broader perspective.
> > >
> > > Once again, we sincerely appreciate the reviewer’s thoughtful engagement. Their constructive feedback and encouraging comments have been invaluable in helping us improve our work.

---

### Decision · Program_Chairs · 2025-05-01

**Decision:**

Accept (poster)

**Comment:**

The authors study distributed optimization on the Stiefel manifold and propose an algorithm that combines the recently introduced Landing method with the well-established error-feedback technique from federated optimization, aimed at reducing communication costs. The paper includes convergence guarantees, a generalization to block-wise constraints, and empirical evaluation. Since the individual algorithmic components are known, the theoretical results are not particularly surprising. Reviewers raised concerns regarding insufficient discussion of related work, unclear technical novelty, lack of justification for certain algorithmic choices, and missing empirical comparisons. In the rebuttal, the authors provided additional empirical results and explanations, which partially addressed these issues. I strongly encourage the authors to thoroughly address these concerns in the next revision.